

# Comprehensive analyses of source sensitivities to and apportionments of PM₂.₅ and ozone over Japan via multiple numerical techniques

Satoru Chatani[1], Hikari Shimadera[2], Syuichi Itahashi[3], Kazuyo Yamaji[4]

[1]National Institute for Environmental Studies, Tsukuba, Ibaraki 305-8506, Japan
[2]Osaka University, Suita, Osaka 565-0871, Japan
[3]Central Research Institute of Electric Power Industry, Abiko, Chiba 270-1194, Japan
[4]Kobe University, Kobe, Hyogo 658-0022, Japan

*Correspondence to*: Satoru Chatani (chatani.satoru@nies.go.jp)

**Abstract.** Source sensitivity and source apportionment are two major indicators representing source-receptor relationships,
which serve as essential information when considering effective strategies to accomplish improved air quality. This study evaluated source sensitivities to and apportionments of ambient ozone and PM₂.₅ concentrations over Japan with multiple numerical techniques embedded in regional chemical transport models, including a brute forth method (BFM), a high-order decoupled direct method (HDDM), and an integrated source apportionment method (ISAM), to update the source-receptor relationships considering stringent emission controls recently implemented in Japan and surrounding countries. We also
attempted to understand the differences among source sensitivities and apportionments calculated by multiple techniques. While domestic sources had certain source apportionments to ozone concentrations, transport from outside Japan dominated the source sensitivities. Although the PM₂.₅ concentrations and absolute magnitudes of their source sensitivities were significantly lower than those reported by previous studies, transport from outside Japan still has relatively large contributions to PM₂.₅ concentrations, implying that there has been a reduction in Japanese emissions, similar to surrounding countries
including China, due to implementation of stringent emission controls. HDDM allowed us to effectively understand the importance of the nonlinear responses of PM₂.₅ concentrations to precursor emissions. Apportionments derived by ISAM were useful in distinguishing various direct and indirect influences on ozone and PM₂.₅ concentrations. It was suggested that that ozone transported from outside Japan plays a key role in exerting various indirect influences on the formation of ozone and secondary PM₂.₅ components. This study demonstrated that a combination of sensitivities and apportionments derived by the
BFM, HDDM, and ISAM can provide critical information to identify key emission sources and processes in the atmosphere, which are vital for the development of effective strategies for improved air quality.

## 1. Introduction

The air quality of Japan has gradually improved. However, ambient concentrations of fine particulate matter smaller than 2.5 micrometres (PM₂.₅) and photochemical oxidants (predominantly ozone) exceed the Environmental Quality Standards
(EQS). Therefore, we must develop effective strategies to suppress ambient PM₂.₅ and ozone concentrations. Quantitative



source-receptor relationships serve as essential information when considering effective strategies. There are two major indicators representing source-receptor relationships (Clappier et al., 2017). One is source sensitivity, which corresponds to a change in ambient pollutant concentrations caused by a certain perturbation in precursor emissions. The second is source apportionment, which corresponds to the contribution of precursor emissions to ambient pollutant concentrations. Receptor
modelling, including Chemical Mass Balance (CMB) and Positive Matrix Factorization (PMF) methods, have been widely applied to evaluate source apportionments (Hopke, 2016). However, they have limitations when attempting to treat secondary pollutants, which form in the atmosphere via complex photochemical reactions. Moreover, receptor modelling cannot evaluate source sensitivities. Forward modelling using a regional chemical transport model is a powerful tool for evaluating both the source sensitivities and apportionments to primary and secondary pollutants.

Several numerical techniques have been developed for regional transport models to evaluate source sensitivities and apportionments (Dunker et al., 2002;Cohan and Napelenok, 2011). A simple technique for evaluating source sensitivities is the brute force method (BFM). Differences in the simulated pollutant concentrations between two simulation cases with and without perturbations in the input precursor emissions is considered as the sensitivity of a given emission source based on the BFM. This technique can require significant computational demand when evaluating the sensitivities of many emission sources.
A decoupled direct method (DDM) is a numerical technique that simultaneously tracks the evolution of sensitivity coefficients, in addition to pollutant concentrations when solving model equations (Yang et al., 1997). This method has been extended to a high-order DDM (HDDM) to track high-order sensitivity coefficients (Hakami et al., 2003). The ozone source apportionment technology (OSAT) (Dunker et al., 2002) and particulate matter source apportionment technology (PSAT) (Wagstrom et al., 2008) are numerical techniques that evaluate the source apportionments of ozone and particulate matter concentrations,
respectively, by tagging contributions of precursor emissions to simulated concentrations. An integrated source apportionment method (ISAM) is a similar numerical technique that evaluates source apportionments (Kwok et al., 2013). Each method has its strengths and weaknesses, such that it is important to appropriately interpret results that will be used to develop effective strategies.

Source sensitivities and apportionments of ambient pollutant concentrations over Japan have been evaluated using
regional chemical transport models. Chatani et al. (2011) evaluated the sensitivities of domestic sources and transboundary transport to simulated $PM_{2.5}$ concentrations over three metropolitan areas in Japan in the 2005 fiscal year. Ikeda et al. (2015) evaluated the sensitivities of source regions in Japan, Korea, and China to simulated $PM_{2.5}$ concentrations over the nine receptor regions in Japan in 2010. These two studies only employed the BFM to derive source sensitivities to $PM_{2.5}$ concentrations. Itahashi et al. (2015) evaluated the sensitivities and apportionments of sources in Japan, Korea, and China to simulated ozone
concentrations over East Asia. This study presented a unique exercise discussing the differences in source sensitivities and apportionments derived by multiple techniques, including the BFM, HDDM, and OSAT, in Asia; these differences have only been discussed in limited studies targeting the United States and Europe (Koo et al., 2009;Burr and Zhang, 2011;Thunis et al., 2019). Expanding targets is key to obtaining a more comprehensive and appropriate understanding of the source sensitivities





and apportionments derived by multiple techniques to pollutant concentrations, including ozone and PM$_{2.5}$, across Asia,
including Japan.

In addition, recent studies (Ronald et al., 2017;Wang et al., 2017;Zheng et al., 2018) suggest that stringent emission controls implemented in China have achieved improved air quality. These improvements should affect air quality not only in China but also across downwind regions including Japan. We must, therefore, update source sensitivities and apportionments when considering additional effective strategies aimed at further air quality improvement in Japan.

Mutual inter-comparisons of the source sensitivities and apportionments derived by multiple models and numerical techniques is one of the objectives of Japan's Study for Reference Air Quality Modelling (J-STREAM) (Chatani et al., 2018b). Model inter-comparisons conducted in earlier phases of J-STREAM have contributed to the derivation of model configurations and development of emission inventories, both of which have contributed to improved model performance (Chatani et al., 2020;Yamaji et al., 2020). As one of the subsequent activities of J-STREAM, this study evaluates the sources sensitivities to
ozone and PM$_{2.5}$ concentrations simulated over regions in Japan for a recent year using the outcomes obtained in earlier phases of J-STREAM. Comprehensive analyses from various perspectives were performed to evaluate the sensitivities of eight domestic and two natural emission source groups, as well as foreign anthropogenic emission sources and transboundary transport throughout the entire 2016 fiscal year. In addition, we perform mutual comparisons of the source sensitivities and apportionments to simulated ozone and PM$_{2.5}$ concentrations. Although the target periods were limited to two weeks in four
seasons, we discuss notable characteristics with respect to the differences in the source sensitivities and apportionments derived by the BFM, HDDM, and ISAM.

## 2. Methodology

### 2.1. Model configuration

The Community Multiscale Air Quality (CMAQ) modelling system (Byun and Schere, 2006) version 5.0.2, in which
both the HDDM and ISAM are embedded, was selected to calculate the source sensitivities and apportionments, in addition to ambient pollutant concentrations. The carbon bond chemical mechanism with the updated toluene chemistry (CB05-TU) (Whitten et al., 2010) and aero6 aerosol module were employed. Input meteorological fields were simulated by the Weather Research and Forecasting (WRF) - Advanced Research WRF (ARW) version 3.7.1 (Skamarock et al., 2008).

Horizontal locations and resolutions of the four target domains, named as d01, d02, d03, and d04, remain unchanged
since the first phase of J-STREAM (Chatani et al., 2018b), as shown in Fig. 1. The top height of the model was lifted from 10,000 to 5,000 Pa to explicitly treat transport in the lower stratosphere (Itahashi et al., 2019a). The vertical layer heights were adjusted to be consistent with those of Chemical Atmospheric Global Climate Model for Studies of Atmospheric Environment and Radiative Forcing (CHASER) (Sudo et al., 2002), which was used to provide boundary concentrations, to avoid numerical diffusions to adjacent layers. Each vertical layer of CHASER from the ground to 80,000 Pa was further divided into two to
simulate vertical variations in the lower atmosphere in more detail. The bottom layer height was approximately 28 m.



Following several changes were applied to the original WRF configuration employed in the first phase of J-STREAM described in Chatani et al. (2018b) based on the outcomes of the model inter-comparisons. The input land use dataset was replaced with one created from geographic information system (GIS) data based on the 6th and 7th Vegetation Surveys released by the Biodiversity Centre of Japan, Ministry of Environment, which yielded improved performance for multiple

meteorological parameters over urban areas (Chatani et al., 2018a). Lakes were added to the dataset based on the National Land Numerical Information Lakes Data. The shortwave and longwave radiation schemes were replaced with the RRTMG schemes (Iacono et al., 2008) to use the climatological ozone and aerosol profiles with spatial, temporal, and compositional variations (Tegen et al., 1997). Microphysics and cumulus schemes had significant influences on the simulated pollutant concentrations in the model inter-comparisons. A Morrison double-moment microphysics scheme (Morrison et al., 2009) and

Grell-Devenyi ensemble cumulus scheme (Grell and Devenyi, 2002) were newly selected because they were characterised by better performance during the sensitivity experiments. Analysis datasets were replaced with the finer ones, i.e., the NCEP GDAS/FNL 0.25 Degree Global Tropospheric Analyses and Forecast Grids (ds083.3) (National Centers for Environmental Prediction/National Weather Service/NOAA/U.S. Department of Commerce, 2015) and Group for High Resolution Sea Surface Temperature (GHRSST) (Martin et al., 2012), for the initial and boundary conditions, as well as grid nudging. Nudging

coefficients are critical parameters for model performance (Spero et al., 2018), but forcing terms in the model equations may disturb physical consistencies. While nudging coefficients for winds were set to $1.0 \times 10^{-4}$ sec$^{-1}$ for all domains and vertical layers, those for temperature and water vapor were reduced to $5.0 \times 10^{-5}$, $3.0 \times 10^{-5}$, $1.0 \times 10^{-5}$, and $1.0 \times 10^{-5}$ sec$^{-1}$ for d01, d02, d03, and d04, respectively. In addition, nudging for the temperature and water vapor within the planetary boundary layer in d03 and d04 was turned off to avoid excessive nudging to finer spatial and temporal scales than the input analysis datasets,

as well as to allow the simulated values to be in accordance with the physical equations.

## 2.2. Emission inputs

Various improvements were applied to the original emission inputs used in the first phase of J-STREAM described in Chatani et al. (2018b) based on the outcomes of the model inter-comparisons. Hemispheric Transport of Air Pollution (HTAP) emissions version 2.2 (Janssens-Maenhout et al., 2015) was used for anthropogenic sources and international shipping

for Asian countries except for Japan. While its target year is 2010, the ratios of sectoral annual emissions reported by Zheng et al. (2018) and the Clean Air Policy Support System (CAPSS) (Lee et al., 2011) were multiplied for China and South Korea, respectively, to represent the changes in the precursor emissions of recent years. Itahashi et al. (2018) suggested the importance of heterogeneous reactions involving Fe and Mn in sulphate formation. The speciation profiles of Fu et al. (2013) were applied to consider other components, including Fe and Mn, in addition to originally available black and organic carbon in $PM_{2.5}$

emissions. The $PM_{2.5}$ emission inventory developed by the Ministry of Environment for the 2015 fiscal year was used for on-road and other transportation sectors in Japan. Emissions from stationary sources in Japan developed in J-STREAM (Chatani et al., 2018b) were fully updated to the 2015 fiscal year with the following improvements. The emission database of large point sources discretized into sectors, facilities, and fuel types were newly developed by Chatani et al. (2019) based on the Research



of Air Pollutant Emissions from Stationary Sources to represent emissions characteristics and speciation profiles including Fe
and Mn. Missing fugitive volatile organic compound (VOC) emission sources, including the use of repellents, air fresheners,
aerosols inhalers, cosmetic products, and products for car washing and repair, were added to be consistent with the Greenhouse
Gas Inventory Office of Japan (2018). $NH_3$ emissions from fertilizer use and manure management were replaced by the values
reported by the Greenhouse Gas Inventory Office of Japan (2018). Fugitive VOC and PM emissions from manure management
were newly estimated based on the European Environment Agency (2016). Emission factors of other $NH_3$ sources, including
human sweat, human breath, dogs, and cats, were replaced by those reported in Sutton et al. (2000). PM emissions from the
abrasion of railways wires and rails were newly estimated as one of the major sources of Fe and Mn. The method to estimate
emissions from open agricultural residue burning were replaced by that used by the Greenhouse Gas Inventory Office of Japan
(2018). We applied the emission factors reported in Fushimi et al. (2017) and Hayashi et al. (2014), as well as the temporal
variations from Tomiyama et al. (2017). Biogenic VOC emissions were estimated by Chatani et al. (2018a) using a detailed
database of vegetation and emission factors specific to Japan. The surf zone, defined as zones adjacent to beaches in the
National Land Numerical Information Land Use Fragmented Mesh Data, was newly added to estimate higher sea salt emissions
from these areas (Gantt et al., 2015) in the CMAQ.

## 2.3. Simulation setup

Ambient pollutant concentrations in d01, d02, d03, and d04 were simulated for the entire 2016 fiscal year (from April
2016 to March 2017). Simulations for the preceding month (March 2016) were treated as spin-up. Sensitivities of the emission
source groups, classified as listed in Table 1, were evaluated by the BFM, in which the emissions of each source group were
reduced by 20% for the entire fiscal year in d02 and two selected weeks in spring (from May 6 to 20), summer (from July 21
to August 4), autumn (from October 20 to November 3), and winter (from January 19 to February 2 of 2017) in d03 and d04.
These two weeks in the four seasons were the periods in which the monitoring campaigns for the ambient concentrations of
the $PM_{2.5}$ components were conducted throughout Japan. The reason for choosing 20% reduction as a perturbation range in
BFM is that it is a typical range of emission reduction by potential emission controls. For s11 (transport through the boundaries
of d02), the boundary concentrations of all species for d02 were reduced by 20%. Differences in the concentrations scaled by
five between the simulations with and without 20% perturbations were treated as sensitivities in this study. In addition, source
sensitivities and apportionments of all the emission source groups listed in Table 1 were evaluated by the HDDM and ISAM,
respectively, using consistent inputs for the two coincident weeks in the four seasons in d02. The first- and second-order
sensitivity coefficients for gaseous precursors of a single emission source group were calculated using HDDM. We note that
the HDDM results were missing for the seasons other than winter because the simulations were not successfully completed
due to numerical convergence problems. Table S1 in the Supplementary Material lists the annual total emission amounts for
each source group in d02.





## 3. Results and discussion

### 3.1. Model performance on ozone and PM$_{2.5}$

We evaluated the model performance for the ozone and PM$_{2.5}$ concentrations in d02 for the entire 2016 fiscal year. Table S2 in the Supplementary Material lists the statistics for the model performance of the maximum daily 8-h average ozone (MDA8O3) and daily mean PM$_{2.5}$ concentrations. Table S2 includes the normalized mean bias (NMB), normalized mean error (NME), and correlation coefficient (R) (Emery et al., 2017) for entire Japan (JP), six regions, and three areas designated by the Automobile NO$_X$-PM law as polluted urban areas. Figure 1 denotes the locations and abbreviations of the six regions and three designated areas. Automatic continuous monitoring data obtained at the ambient air pollution monitoring stations (APMSs) were used. Figure S1 in the Supplementary Material compares the observed and simulated monthly mean MDA8O3 and PM$_{2.5}$ concentrations averaged at all stations in the regions.

The MDA8O3 were slightly overestimated in all regions. The observed MDA8O3 was the highest in May and lowest in December. There was another peak in August in western Japan. The model consistently reproduced these monthly variations. Overestimation occurred from the peak in May to the valley in December. Values from December to March were slightly underestimated. The overestimation in summer in this study is less evident than that reported in Chatani et al. (2020), who summarized the performance of the models that participated in the model inter-comparisons conducted in the first phase of J-STREAM. The improved performance obtained in this study may be due to the various improvements in the configurations described in Section 2, as well as differences in the meteorological conditions. Kitayama et al. (2019) shows that CB05-TU, which was employed in this study, tends to yield lower ozone concentrations among major chemical mechanisms. All the criteria proposed by Emery et al. (2017) were attained in all regions.

The PM$_{2.5}$ concentrations were underestimated in all regions. The statistics tended to be worse in eastern Japan as opposed to western Japan. The observed PM$_{2.5}$ concentrations fluctuated with a peak in May and valley near September. Although the simulations reproduced these monthly variations, the absolute values were consistently underestimated. The criteria proposed by Emery et al. (2017) were attained for NME and R, but not for NMB due to persistent underestimation.

As mentioned in Section 2, monitoring campaigns for the ambient concentrations of the PM$_{2.5}$ components were conducted throughout Japan for the two target weeks in spring, summer, autumn, and winter. The components of the particulates collected on filters for throughout one day were analysed. These data are useful for the further validation of model performance for the PM$_{2.5}$ components. Figure S2 in the Supplementary Material shows scatter plots of the observed and simulated daily concentrations of the PM$_{2.5}$ components (SO$_4^{2-}$, NO$_3^-$, NH$_4^+$, elemental carbon (EC), and organic carbon (OC)) at all locations throughout Japan during the monitoring campaigns in all four seasons. Table S1 summarizes their statistics for entire Japan and the four seasons. The simulated average concentrations of SO$_4^{2-}$ and NH$_4^+$ are similar to the observed values. Their observed and simulated values have significant correlations with R, i.e., approximately 0.7. The NO$_3^-$ concentrations were overestimated with NME of over 100%. The R between the observed and simulated values is 0.441, which is significantly lower than SO$_4^{2-}$ and NH$_4^+$. A number of biased dots for NO$_3^-$ occur in the scatter plot. While excessively higher simulated





values appeared in summer, the model underestimated several of the higher values mainly observed in winter. Although previous studies have discussed issues of poor model performance associated with reproducing the $NO_3^-$ concentrations in Japan (Shimadera et al., 2014;Shimadera et al., 2018), they have not yet been solved even after the application of various improvements. Both the EC and OC concentrations were underestimated. As OC is the second major component of $PM_{2.5}$, its underestimation is one of the major causes of $PM_{2.5}$ underestimation. Less overestimated dots are found in their scatter plots, indicating their persistent underestimation. Shimadera et al. (2018) also discussed the issues of poor model performance associated with reproducing OC concentrations in Japan, suggesting condensable organic matter as a key factor for this poor performance. Although studies on this issue have been conducted by Morino et al. (2018), they remain unsolved.

We note that it is important to recognize that source sensitivities and apportionments introduced in the subsequent sections may be affected by the model performance described in this section.

### 3.2. Source sensitivities on the annual mean ozone and PM2.5

Figure 2 shows the source sensitivities to the annual mean ozone and $PM_{2.5}$ concentrations derived by the BFM in all regions. The sensitivity of s11 (transport through the boundaries of d02) is overwhelming to ozone. The sensitivities of domestic sources, including s01 (on-road vehicles) and s04 (stationary combustion), are negative in the three designated areas, which is caused by the titration of ozone due to higher $NO_X$ emissions in urban areas. While s11 has the highest sensitivities to $PM_{2.5}$, those of domestic anthropogenic sources, including s01, s02 (ships), s04, and s08 (agriculture and fugitive ammonia), are also evident. The sensitivities of domestic anthropogenic sources are higher in the three designated areas with higher precursor emissions. Sums of the sensitivities of all the source groups to ozone and $PM_{2.5}$ are lower and higher than their simulated concentrations, respectively, due to the nonlinear relationships between their concentrations and precursor emissions.

The sensitivities to $PM_{2.5}$ reflect the characteristics of the sensitivities to individual $PM_{2.5}$ components. Figure S3 in the Supplementary Material shows the source sensitivities to the annual mean concentrations of the $PM_{2.5}$ components derived by the BFM in all regions. The EC and primary organic aerosol (POA) are primary components. Sums of the sensitivities of all the source groups to these primary components are consistent with the simulated concentrations. Higher sensitivities for specific source groups, including s03 (non-road transport) and s05 (biomass combustion), to EC and POA, respectively, are evident in the three designated areas. Sums of the sensitivities of all the source groups to $SO_4^{2-}$, which is mainly a secondary component but almost non-volatile, are also equivalent to the simulated concentrations. The sensitivities of s09 (natural) are higher in western Japan, i.e., the location of several active volcanoes. Significant nonlinearities exist in the sensitivities to $NO_3^-$ and $NH_4^+$, which are mainly secondary components. Specifically, although s08 mainly emits $NH_3$ and no $NO_X$, it is highly sensitive to $NO_3^-$ concentrations due to indirect influences. Details of these nonlinearities are discussed in section 3.6, which compares the source sensitivities and apportionments. The sensitivities of s12 (sea salt) are negative to $NO_3^-$ and $NH_4^+$. $Cl^-$ originated from sea salts and mostly involved in coarse particles tend to be replaced by $NO_3^-$ due to chlorine loss (Pio and Lopes, 1998;Chen et al., 2016). If sea salts are absent, $NO_3^-$ is more likely to be kept in $PM_{2.5}$ with $NH_4^+$. Nonlinearities are



also significant to secondary organic aerosol (SOA). Biogenic VOC emissions included in s09 are specifically sensitive to SOA.

Table S3 in Supplementary Material lists the ratios (normalized or not) of the source sensitivities to the annual mean ozone and PM$_{2.5}$ concentrations simulated in the regions, which were compared with previous studies. The annual mean PM$_{2.5}$ concentrations simulated in this study for the three designated areas are 6–9 μg/m$^3$, which is significantly lower than

approximately 16 μg/m$^3$ simulated by Chatani et al. (2011) for the corresponding areas in the 2005 fiscal year. However, their ratios for the sensitivities of foreign anthropogenic sources are 48, 41, and 31% in OH, AM, and ST, respectively, which are lower than the approximately 65% calculated in this study as the sums of the sensitivities for s10 (anthropogenic in other countries in d02) and s11. The normalized ratios for the sensitivities of the sources in Korea and China for 2010 were 71, 57, and 39% in Kyushu, Kinki, and Kanto, respectively, reported in Ikeda et al. (2015), whereas the sensitivities of s10 and s11 in

KO, KS, and KK, which are equivalent to Kyushu, Kinki, and Kanto, were 68, 65, and 59%, respectively, in this study. Relative contributions of foreign sources evaluated in this study are higher than previous studies for most areas of Japan despite the stringent emission controls implemented in China. There are two possible reasons for these elevated contributions. Zheng et al. (2018) showed that the emissions of PM$_{2.5}$, SO$_2$, and NO$_X$ in China decreased by 31, 52, and 15%, respectively, from the 2010 to 2016 due to stringent emission controls. If we compare the emissions reported in Chatani et al. (2011) with those used

in this study, the emissions of PM$_{2.5}$, SO$_2$, and NO$_X$ in Japan decreased by 29, 48, and 33%, respectively, from fiscal years 2005 to 2015. Therefore, the relative emission reductions in Japan may be larger than those in China if we assume certain changes in the emissions from 2005 to 2010. In particular, stringent emission controls implemented on diesel vehicles by the central and local governments were quite effective in suppressing PM$_{2.5}$ emissions and ambient concentrations in urban areas (Kondo et al., 2012). A reduction in the activity of the Miyakejima volcano in recent years has also resulted in lower SO$_2$

emissions. We can also state that the underestimations of the PM$_{2.5}$ concentrations are larger in eastern than western Japan as described in section 3.1. Less contrasts in the sensitivities of foreign sources evaluated in this study between western Japan, which is more affected by westerly wind transport, and eastern Japan imply excessive estimates of emission reductions in Japan. Another reason is the fact that s11 includes all the components that pass through the boundaries of d02, such that it is affected not only by anthropogenic sources in China but also anthropogenic sources in other countries, natural sources, and

background concentrations.

Ozone concentrations have been nearly stable in Japan in recent years while the NO$_X$ and VOC concentrations have been supressed (Wakamatsu et al., 2013). Sensitivities derived in this study suggest that a continuous reduction in the NO$_X$ emissions, due to stringent emission controls implemented in Japan, has resulted in increases in the annual mean ozone concentrations caused by less titration of the ozone in urban areas. Suppressing the annual mean ozone concentrations further

is difficult because domestic sources are practically insensitive. Trends in the transboundary transport of ozone likely have a significant effect on the mean annual ozone concentrations (Kurokawa et al., 2009;Chatani and Sudo, 2011). In contrast, domestic sources, as well as transport from outside Japan, are sensitive to the annual mean PM$_{2.5}$ concentrations. The stringent emission controls implemented in Japan and surrounding countries appear to have contributed to their decreasing trends in



Japan. Additional efforts to reduce emissions may produce further improvements in the annual mean PM$_{2.5}$ concentrations,
whereas further validations of the emissions in Japan are necessary.

### 3.3. Monthly variations in source sensitivities on ozone and PM$_{2.5}$

Figure 3 shows the source sensitivities to the monthly mean ozone and PM$_{2.5}$ concentrations derived by BFM simulated in entire Japan (JP) and ST, which is one of the three designated areas, including the Tokyo metropolitan area. Figure S4 in the Supplementary material shows the sensitives to the PM$_{2.5}$ components. Negative sensitivities of the domestic sources,
including s01 (on-road vehicles) and s04 (stationary combustion), to the ozone concentrations are evident in winter due to the titration of ozone by higher NO$_X$ emissions and inactive photochemical reactions in urban areas. The sensitivity of s11 (transport through the boundaries of d02) is higher than the simulated concentrations, indicating that more ozone is transported from the outside and titrated by NO$_X$ emissions in Japan. In contrast, negative sensitivities of domestic sources are less evident in summer even in the ST. Reductions in the ozone by titration are compensated by ozone formation from precursor emissions
originating from domestic sources due to more active photochemical reactions. Differences can be observed in the major source groups, which have positive sensitivities in summer in JP and ST. While the sensitivities of s02 (ships) and s04, which mainly emit NO$_X$, are higher in JP, those of s07 (fugitive VOC) and s09 (natural), which mainly emit VOC, are higher in ST.

The sensitivity of s11 to PM$_{2.5}$ is the highest in May due to transport by dominant westerly winds in this season. The sensitivity to POA is predominantly high, suggesting that it is affected by variable sources, such as open biomass burning. The
sensitivities of s02, s04, and s09, which are mainly located in the southern sides of Japan, are higher in the summer, caused by dominant southerly winds, as well as active secondary formation, which are clearly reflected in their sensitivities to SO$_4^{2-}$. The sensitivities of s01 and s08 (agriculture and fugitive ammonia) are high in winter. A colder and more stable atmosphere in winter favours the accumulation of emissions from local sources and the partitioning of NO$_3^-$ and NH$_4^+$ to the aerosol phase, as reflected in their sensitivities.

As discussed for the annual mean concentrations, suppressing the monthly mean ozone concentrations is difficult because the sensitivities of s11 are overwhelming in all months. In particular, the sensitivities of s01 and s04 are negatively large in urban areas in autumn and winter. Further reductions in their NO$_X$ emissions may result in additional increases in the monthly mean ozone concentrations in these seasons. We note that the ozone concentrations simulated in these seasons are lower than other seasons. Returning to the background concentration levels via reduced titration is inevitable. In contrast, the
negative sensitivities are less evident in spring and summer. Reductions in the precursor emissions for certain domestic sources have the possibility to suppress, to a certain extent, the monthly mean ozone concentrations. Effective sources may be different in urban and other areas due to differences in ozone formation regimes (Inoue et al., 2019). The effects that strategies have on various sources of precursor emissions for PM$_{2.5}$ may vary seasonally due to differences in meteorological and photochemical conditions.



### 3.4. Source sensitivities per unit precursor emissions

Air quality standards are defined in terms of ambient concentrations while targets for emission controls are defined in terms of emission amounts. Therefore, understanding if the sensitivities per equal emission amounts to ambient concentrations are consistent for different sources is important. Figure 4 shows the sensitivities per annual total amount of precursor emissions for domestic anthropogenic sources (s01–s08) in d02 to the annual mean ambient concentrations of the corresponding PM$_{2.5}$ components in all of Japan. The horizontal and vertical locations of the emissions have an effect on the differences in the values for the primary components (EC and POA). Here, s02 includes ship emissions in surrounding oceans in d02, whose values suggest that approximately 40% of the ship emissions in d02 affect the concentrations of primary PM$_{2.5}$ components over Japan. The values of s03 (non-road transport) and s04 (stationary combustion) are slightly lower because they include elevated sources, such as airplanes and large point sources. Slight differences among s01 (on-road vehicles), s05 (biomass combustion), and s06 (residential combustion), whose emissions were ingested only in the bottom layer, may be caused by differences in their horizontal distributions. Sources located in coastal areas may have lower influences as their emissions are transported beyond the land. Additional differences caused by photochemical reactions were observed for secondary components. The value of s05 include agricultural residue burning, which has large spatial and temporal variations, such that its emissions may be high where secondary formation is relatively active. The value for NH$_4^+$ in s01 is significantly higher than that of s08 (agriculture and fugitive ammonia) because s01 co-emits NO$_X$ and NH$_3$, which have a mutual correlation.

The effectiveness of equal reduction amounts of the precursor emissions may be different among sources due to photochemical reactions, as well as the locations of emissions, which are factors that may need to be considered when exploring effective strategies.

### 3.5. Differences in source sensitivities among domains

Nesting is a technique in air quality simulations aimed at obtaining improved model performance using finer meshes over target regions, as well as representing large-scale transport in coarser meshes in a computationally effective manner. This study employed d03 and d04 with finer 5 x 5 km meshes over OH, AM, and ST, which are all the major target urban areas. We emphasize the importance of observing how much the sensitivities evaluated in d03 and d04 are different from those in d02 using coarser 15 x 15 km meshes. Figure 5 shows the sensitivities of all the source groups over OH, AM, and ST evaluated in d02, d03, and d04 averaged for the two target weeks during the four seasons. The ozone concentrations simulated for the summer in d02 and d03 or d04 are slightly different. Negative sensitivities of s01 (on-road vehicles) and s04 (stationary combustion) are correspondingly higher. Finer meshes tend to result in slightly larger influences of ozone titration. While the simulated PM$_{2.5}$ concentrations are slightly different in different domains, the relative contributions of the source groups to the sensitivities are consistent. These results suggest that differences in horizontal resolutions between d02 and d03 or d04 do not cause critical differences in the sensitivities when they are spatially and temporally averaged over the target areas and two weeks. They also support the validity of the discussions in this study, which are mostly based on the results obtained in d02.





### 3.6. Mutual comparisons of source sensitivities and apportionments derived by BFM, HDDM, and ISAM

#### 3.6.1. Overall differences among techniques

Figure 6 shows the apportionments derived by ISAM and sensitivities derived by BFM and HDDM of all the source groups to the simulated ozone and $PM_{2.5}$ concentrations in entire Japan (JP) and ST averaged for the two target weeks during the four seasons. We used the following treatments in Fig. 6. Only the sensitivities of the gaseous precursor emissions were calculated by HDDM. The sensitivities of emissions and boundary concentrations of primary aerosol components (EC, POA, and other primary components) calculated by BFM were also used for HDDM. Apportionments to SOA concentrations were not calculated by ISAM. The simulated SOA concentrations were characterised as apportionments of "OTHR" in ISAM. The HDDM sensitivities were evaluated using first- and second-order sensitivity coefficients ($S^{(1)}$ and $S^{(2)}$) based on the following Taylor expansion (Eq. (1)):

$$C(+\Delta\varepsilon) = C(0) + \Delta\varepsilon S^{(1)}(0) + \frac{\Delta\varepsilon^2}{2}S^{(2)}(0), \tag{1}$$

where $C(+\Delta\varepsilon)$ and $C(0)$ are the simulated concentrations with and without the perturbations, respectively, $\Delta\varepsilon$ is a perturbation ratio, and $S^{(1)}(0)$ and $S^{(2)}(0)$ are the first- and second-order sensitivity coefficients, respectively. The HDDM-20 corresponds to the value calculated by applying $\Delta\varepsilon = -0.2$ and multiplication by 5, which is equivalent to the value obtained by the BFM. The HDDM-100 corresponds to the value calculated by applying $\Delta\varepsilon = -1.0$. Differences between HDDM-20 and HDDM-100 correspond to the influences of nonlinear responses against changes in the precursor emissions. Sums of the apportionments of all the source groups derived by ISAM represent, in principle, the simulated concentrations.

Not only the sensitivities described in previous sections but also the apportionments of s11 (transport through the boundaries of d02) to ozone concentrations are overwhelming, suggesting that ozone over Japan is dominantly transported from outside Japan. Domestic sources, including s01 (on-road vehicles), s02 (ships), and s04 (stationary combustion), have certain positive apportionments to ozone concentrations in the spring and summer, indicating that a certain amount of ozone forms from precursors emitted from these sources. Nevertheless, domestic sources have small or even negative sensitivities to ozone concentrations. Let us consider a simple example. Ozone transported from outside Japan reacts with NO emitted in Japan and forms $NO_2$ (step 1). Next, $NO_2$ is photochemically decomposed to NO and O, followed by ozone regeneration via a rapid reaction between O and $O_2$ (step 2). Potential ozone (ozone + $NO_2$) is preserved in these two steps (Itahashi et al., 2015). Regenerated ozone is apportioned to NO sources in Japan by ISAM in this case. However, if ozone transported from outside Japan increases and enough NO is available, there is a subsequent equivalent increase in $NO_2$ formation and ozone regeneration. This indicates that regenerated ozone is sensitive to transport from outside Japan. In contrast, if NO emissions in Japan increase, ozone concentrations decrease after step 1 or remain unchanged after step 2. This suggests that the sensitivities of NO sources in Japan are negative after step 1 or none after step 2. Their sensitivities cannot become positive in this example. In reality, a certain amount of the NO is oxidized by other species, including $RO_2$ that originates from VOCs emitted in Japan. They result in net ozone formation and positive sensitivities, which compensates, to a certain extent, negative sensitivities. The





apportionment of s11 to ozone concentrations is smaller than their sensitivities in autumn and winter in ST. The apportionments
355   of domestic sources are negligible in these seasons. Ozone is titrated by high NO emissions in urban areas in step 1 while step
2 is not fully reached due to inactive photochemical reactions.

There are differences in the source apportionments and sensitivities to $PM_{2.5}$, which reflect those to the concentrations
of the $PM_{2.5}$ components, shown in Fig. S5 in the Supplementary Material. Sensitivities to gaseous $HNO_3$ and $NH_3$
concentrations, which are counterparts of $NO_3^-$ and $NH_4^+$ in the gas phase, are also shown in Fig. S5. The source
apportionments and sensitivities to primary components (EC and POA) are consistent. While the sums of the source
apportionments and sensitivities of all the sources to $SO_4^{2-}$ are also consistent, there are differences in the relative contributions
of the source groups. The apportionment of s11 corresponds to the concentrations of $SO_4^{2-}$ transported from outside Japan.
The higher sensitivities are affected by additional indirect influences, i.e., $SO_2$ is oxidized to $H_2SO_4$ via gaseous and aqueous
reactions and is then predominantly partitioned to $SO_4^{2-}$. Gaseous oxidation of $SO_2$ occurs due to OH, a part of which originates
in ozone. Therefore, s11, which has an overwhelmingly high sensitivity to ozone, also has higher sensitivities to $SO_4^{2-}$ oxidized
from $SO_2$. In contrast, if $SO_2$ emissions are reduced under fixed OH, other $SO_2$ remaining in the atmosphere has the opportunity
to be oxidized to $SO_4^{2-}$. Therefore, the sensitivities of downwind domestic sources are smaller than their apportionments.
Similar discussions are applicable to $NO_3^-$. The apportionments of s11 to $NO_3^-$ and $HNO_3$ are lower than its sensitivities,
indicating that a certain amount of the $NO_3^-$ and $HNO_3$ is not directly transported from outside Japan. Ozone overwhelmingly
affected by s11 enhances the oxidation of $NO_X$ to $HNO_3$ through OH, followed by a smaller amount that is further partitioned
to $NO_3^-$. This causes indirect influences on the sensitivities of s11. Such influences are apparent in the horizontal distributions
of the apportionments and sensitivities of s11 to concentrations of related species for the two target weeks of spring shown in
Fig. S6 in the Supplementary Material. The sensitivities to $SO_4^{2-}$ and $NO_3^-$ are higher than their apportionments over Japan.
The sensitivities of $SO_2$ and $NO_2$ over Japan are correspondingly negative, suggesting that they are oxidized by OH that
originated in ozone transported from outside Japan. The isolated higher sensitivities over Japan, particularly visible for those
to $NO_3^-$, clearly suggest that they are not directly transported from outside Japan.

Section 3.2 discussed higher relative contributions than previous studies and less contrasts between western and
eastern Japan for the sensitivities of s11 to the $PM_{2.5}$ concentrations obtained in this study. Oxidation of precursors emitted
from domestic sources by OH that originated in ozone transported from outside Japan is another factor that causes higher
sensitivities of s11. The entirety of Japan is equally affected by ozone transported from outside Japan, as shown in Fig. 2(a),
due to its long lifetime in the atmosphere, resulting in less contrasts in the sensitivities of s11 between western and eastern
Japan. Ozone governs an oxidative capacity of the atmosphere (Prinn, 2003). If ozone transported from outside Japan is not as
reduced in future, efforts to reduce precursor emissions in Japan will not effectively contribute to the reduction in the
concentrations of secondary $PM_{2.5}$ components because OH that originated in ozone transported from outside Japan affects
their formation.

s08 (agriculture and fugitive ammonia), which emits $NH_3$ and no $NO_X$, has no apportionments to $NO_3^-$ in accordance
to the principle. Nevertheless, it has high sensitivities to $NO_3^-$, which are affected by the relationships between $NH_4^+$ and $NO_3^-$.





Here, $NH_4^+$ and $NO_3^-$ are mutual counter ions in $NH_4NO_3$, whose formation is enhanced when both are available. More $NH_3$ emissions can induce the partitioning of $HNO_3$ to $NH_4NO_3$. These influences can be observed in the correspondingly negative

sensitivities of s08 to gaseous $HNO_3$. While the apportionments to $NH_4^+$ are dominated by s08 in ST, its sensitivities are significantly smaller than the apportionments. Both $(NH_4)_2SO_4$ and $NH_4NO_3$ are major forms of $NH_4^+$, where, as discussed above, $NH_4NO_3$ formation is sensitive to $NH_3$ emissions. In contrast, the sensitivities of s08 to $SO_4^{2-}$ concentrations are negligible (Fig. S5(c2)), suggesting that $(NH_4)_2SO_4$ formation is dominantly limited by $SO_2$ sources, including s02 and s04. Their influences are reflected in the sensitivities of s02 and s04 to $NH_4^+$ concentrations (Fig. S5(e2)). These results are

consistent with Clappier et al. (2017), who discuss the differences between apportionments and sensitivities in different regimes involving $SO_2$, $NO_X$, and $NH_3$ using idealized example cases.

s08 has certain sensitivities to $PM_{2.5}$ concentrations, as shown in Fig. 2(b), which are indirectly caused by the interactions between $NH_4^+$ and $NO_3^-$. There are several studies that insist on the importance of $NH_3$ emission controls to reduce $PM_{2.5}$ concentrations (Pinder et al., 2007;Wu et al., 2016;Guo et al., 2018). Such discussions are applicable to Japan. However,

Liu et al. (2019) suggested that $NH_3$ emission control could worsen acid rain because nitric acid is not neutralized and remains in the atmosphere. Not only reducing $PM_{2.5}$ concentrations, but also other environmental aspects, including acid rain and nitrogen cycles, should be considered to develop strategies effective at accomplishing sustainable developments.

### 3.6.2. Nonlinear responses in sensitivities

Differences among the sensitivities derived by BFM, HDDM-20, and HDDM-100 are mostly small, suggesting that,

in most cases, HDDM is able to calculate sensitivities consistent with BFM. Slight differences were found in the sensitivities to $NO_3^-$ concentrations derived by them. Figure 7(a) shows the sensitivities of the source groups located within d02 (s01–s10) derived by BFM, HDDM-20, and HDDM-100 to the daily $NO_3^-$ concentrations for the two target weeks in winter in ST. The sensitivities derived by BFM are slightly higher than those derived by HDDM-20. While the sensitivities derived by HDDM-20 are only affected by gaseous precursor emissions, those derived by BFM contain minor contributions of primary emitted

$NO_3^-$. They are one of the factors that may result in higher sensitivities derived by BFM. However, differences were found even in the sensitivities of s08 (agriculture and fugitive ammonia), which mostly emits $NH_3$. Differences should be recognized as uncertainties that originated from two principally different methodologies.

Sums of the sensitivities derived by HDDM-100 are higher than those derived by HDDM-20 for all days, indicating nonlinear responses of $NO_3^-$ concentrations against precursor emissions. Daily variations of two additional indicators are

shown in Fig. 7(b). One is a nonlinear index (Cohan et al., 2005), which is calculated as follows:

$$Nonlinear\ index = \left| \frac{0.5S^{(2)}}{S^{(1)}} \right|. \tag{2}$$

This corresponds to an absolute ratio of the second- to first-order sensitivity terms when a perturbation is $\Delta \varepsilon = \pm 1.0$, indicating the strength of the nonlinearities. Another indicator is an available $NH_3$ ratio, which corresponds to a ratio of $NH_3$ + $NH_4^+$ (those stoichiometrically equivalent to $SO_4^{2-}$ are subtracted) to $HNO_3$ + $NO_3^-$, indicating an abundance of potential





$NH_4^+$ that can be combined with $NO_3^-$. Here, s08 has the highest nonlinear indices that cause the overall nonlinearities, implying that the $NO_3^-$ concentrations have nonlinear responses to $NH_3$ emissions. Daily variations in the nonlinear indices of s08 and available $NH_3$ ratios are well correlated; nonlinearities are higher when available $NH_3$ ratios are lower. The formation of $NH_4NO_3$ tends to be more constrained by $NH_3$ with less available $NH_3$, as shown by Xing et al. (2011). A typical situation occurred on January 30th. Negative sensitivities of s04 (stationary combustion) suggest that $SO_2$ emissions of s04 remove $NH_3$

to form $(NH_4)_2SO_4$ and prevent $NH_4NO_3$ formation. The HDDM can represent such complex nonlinear relationships involving multiple species.

### 3.6.3. Dependence of ozone formation on $NO_X$ and VOC

ISAM has the capability to separately calculate apportionments of $NO_X$ and VOC emissions of a given source to ozone concentrations based on ozone formation conditions (Kwok et al., 2013). Understanding their relationships with the

corresponding sensitivities is important. Additional simulations were conducted to separately derive the sensitivities of $NO_X$ and VOC emissions of s01 (on-road vehicles) to ozone concentrations by BFM.

Figure 8 shows the apportionments and sensitivities of the $NO_X$ and VOC emissions of s01 derived by ISAM and BFM to daily ozone concentrations for the two target weeks in summer in ST. The apportionment of $NO_X$ emissions is higher than that of the VOC emissions. While there are differences in the magnitudes of the apportionments and sensitivities of the

VOC emissions, they have consistent daily variations. The sensitivity of the $NO_X$ emissions is mostly negative, but became positive on July 25th when the apportionments, as well as the ozone concentrations, were the highest. The dominant winds were northerly until July 24th and switched to southerly on July 25th. Precursors and the ozone formed from them were transported to the south and returned to ST. Therefore, a relatively aged airmass passed over ST on July 25th. Influences of ozone formation from $NO_X$ emissions were higher than the immediate titration by them for this condition.

Figure S7 in the Supplementary Material shows the apportionments and sensitivities of the $NO_X$ and VOC emissions of s01 derived by ISAM and BFM to the hourly ozone concentrations on July 25th in ST. Hourly variations in the apportionments and sensitivities of the VOC emissions are consistent. While the sensitivities of the $NO_X$ emissions during the night are slightly negative due to titration, their higher positive sensitivities during the daytime indicate the contribution of the $NO_X$ emissions to the high ozone concentrations. We note that the sensitivities of the $NO_X$ and VOC emissions calculated by

BFM with 20% perturbations are still lower than their apportionments in this condition, which may be caused by nonlinear responses of ozone concentrations against precursor emissions. Cohan et al. (2005) reported that the sensitivities of ozone concentrations are lower when perturbations of precursor emissions are smaller because other remaining precursors are more likely to contribute to ozone formation instead. This may also be the reason why the sums of the sensitivities of all the sources are lower than the simulated ozone concentrations in spring and summer, as shown in Figs. 2, 3, and 5.

Figure S8 in the Supplementary Material shows the horizontal distributions of the apportionments and sensitivities of the s01 $NO_X$ and VOC emissions to the ozone concentrations averaged for the two target weeks in summer. The sensitivity of $NO_X$ emissions is negative in urban and coastal areas where $NO_X$ emissions from on-road vehicles are high. There are





consistencies in the horizontal distributions of the positive sensitivities and apportionments of $NO_X$ and VOC emissions. While there are quantitative differences in the magnitudes of the sensitivities and apportionments due to nonlinear influences, ISAM

can provide the spatial and temporal variations in the apportionments of $NO_X$ and VOC emissions consistent with the sensitivities derived by BFM.

## 4. Conclusions

Sensitivities and apportionments of emissions from twelve source groups to ozone and $PM_{2.5}$ concentrations over regions in Japan for the 2016 fiscal year were evaluated by the BFM, HDDM, and ISAM using emissions that take into account

the latest stringent emission controls. Transport from outside Japan dominated the source sensitivities to ozone concentrations. While $PM_{2.5}$ concentrations and their absolute sensitivities of all the sources were lower than those calculated by previous studies for past years due to emission reductions, the relative contributions of transport from outside Japan to the total sensitivities were even larger, suggesting that emissions in Japan have been reduced similar to surrounding countries, including China. Moreover, their sensitivities calculated in this study include indirect influences of ozone predominantly transported

from outside Japan via the oxidation of precursors by OH to secondary $PM_{2.5}$ components. Domestic sources had certain sensitivities to $PM_{2.5}$, but significantly smaller or even negative sensitivities to ozone due to titration and nonlinear responses against precursor emissions.

Sensitivities and apportionments for primary species were consistent. Fundamental differences were found between them for secondary species. While apportionments represent direct contributions, sensitivities include indirect influences.

Comparisons between apportionments and sensitivities can help distinguishing direct and indirect influences. Various indirect influences on ozone and $PM_{2.5}$ were identified in this study, including the titration of ozone by $NO_X$ emissions, oxidation of precursors by OH that originated in ozone, and inter-correlations between $NH_4^+$ and $NO_3^-$ in their partitioning, all of which could occur everywhere in the world. Sensitivities derived by BFM and HDDM were mostly consistent. The HDDM also revealed possibilities to represent nonlinear responses of concentrations to precursor emissions. The dependence of ozone

formation on the $NO_X$ and VOC emissions derived by ISAM was spatially and temporally consistent with sensitivities derived by BFM.

Quantitative source-receptor relationships serve as essential information when considering effective strategies, as described in the introduction. Clappier et al. (2017) and Thunis et al. (2019) suggested that sensitivities can provide more useful information than apportionments when considering effective strategies. This study indicates that apportionments

simultaneously evaluated with sensitivities are also helpful to distinguish direct and indirect influences, i.e., they cannot be distinguished only by sensitivities. Understanding the influences that various factors have on sensitivities can contribute to the establishment of effective strategies. However, accurate sensitivities and apportionments depend on model performance. Uncertainties remain in model performance, as discussed in section 3.1. If specific emission sources affect overall model performance, source sensitivities and apportionments derived by models may be skewed.  More studies are necessary to obtain



more confidence in model performance. In addition, sensitivities obtained by the BFM with a single perturbation may be inappropriate for applications to different perturbation ranges when nonlinearities are higher. High-order sensitivity coefficients calculated by the HDDM could help evaluate the importance of nonlinear responses.

This study demonstrated that a combination of sensitivities and apportionments derived by the BFM, HDDM, and ISAM can provide critical information to identify key emission sources and processes in the atmosphere, which are vital for

the development of effective strategies for improved air quality, using consistent model configurations and inputs. In reality, different model configurations and inputs may be used to consider strategies. Itahashi et al. (2019b) reported that source sensitivities can be changed by the regional chemical transport model with improved treatments for aqueous reactions. Uncertainties in the sensitivities and apportionments caused by different model configurations and inputs should be explored as the next step of J-STREAM.

## Data availability


The input datasets are available upon request at http://www.nies.go.jp/chiiki/jstream.html. The output datasets are available upon request to the authors.

## Author contribution

SC designed this study, conducted BFM simulations, and prepared the manuscript. HS and SI conducted ISAM and HDDM

simulations, respectively. KY prepared the meteorology and additional inputs.

## Competing interests

There are no conflicts of interest to declare.

## Acknowledgements

This study was supported by the Environment Research and Technology Development Fund (5-1601 and 5-1802) of

the Environmental Restoration and Conservation Agency of Japan. The data based on the 6th and 7th Vegetation Surveys was obtained from the Biodiversity Centre of Japan, Ministry of Environment (http://gis.biodic.go.jp/webgis/sc-006.html). The National Land Numerical Information data was obtained from the National Land Numerical Information download service (http://nlftp.mlit.go.jp/ksj/index.html). Data from the Research of Air Pollutant Emissions from Stationary Sources was provided by the Ministry of Environment. Automatic continuous monitoring data of the APMSs was obtained from the

National Institute for Environmental Studies (http://www.nies.go.jp/igreen/). The data associated with the monitoring



campaigns for the ambient concentrations of the PM$_{2.5}$ components was obtained from the Ministry of Environment (http://www.env.go.jp/air/osen/pm/monitoring.html).

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



**Table 1. Emission source groups whose sensitivities and apportionments were evaluated in this study.**

| Group | Included emission sources |
|-------|----------------------------|
| s01 | On-road vehicles |
| s02 | Ships |
| s03 | Non-road transport (machineries, railways, and airplanes) |
| s04 | Stationary combustion (power plants, industries, and commercial) |
| s05 | Biomass combustion (smoking, cooking, and agricultural residue burning) |
| s06 | Residential combustion |
| s07 | Fugitive volatile organic compounds |
| s08 | Agriculture (except for agricultural residue burning) and fugitive ammonia |
| s09 | Natural (volcanoes, biogenic, and soil) |
| s10 | Anthropogenic sources in other countries in d02 |
| s11 | Transport through boundaries of d02 |
| s12 | Sea salt |






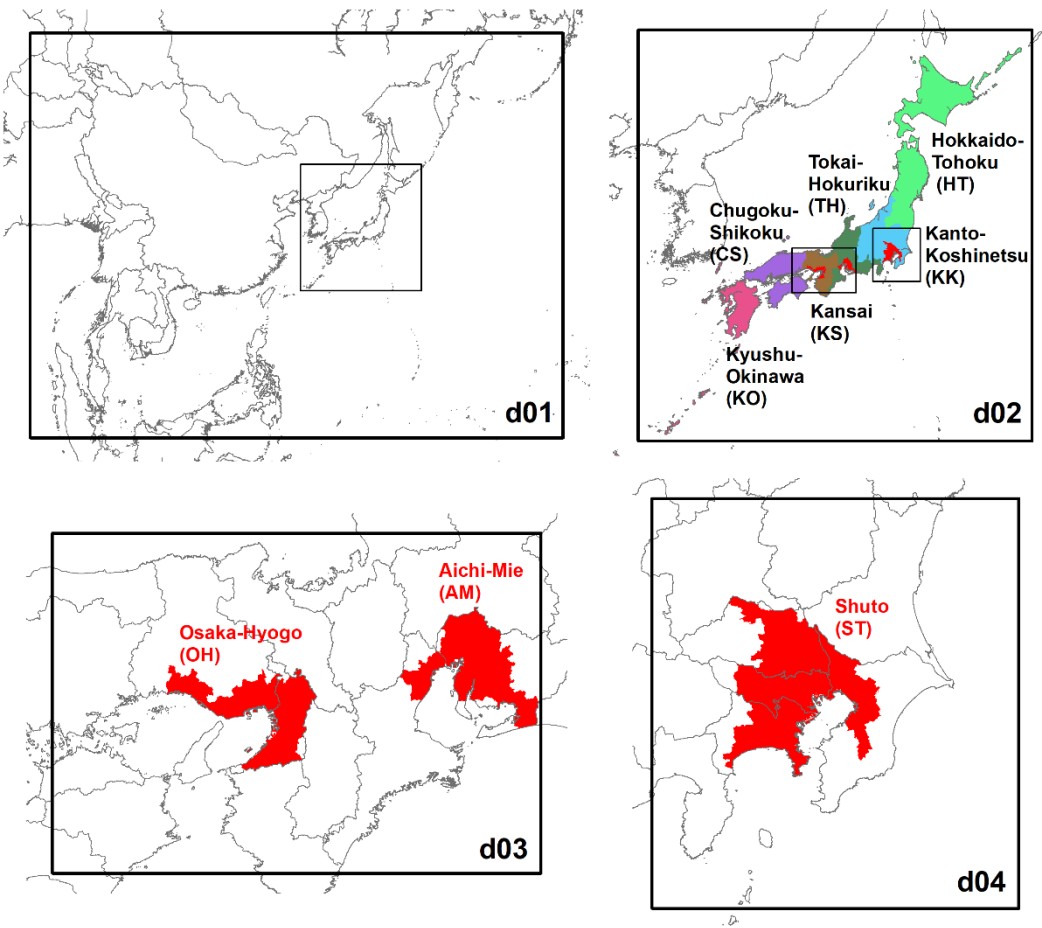

**Figure 1. Target domains for the simulations in this study. Results are summarized for six colour coded regions in d02 and three designated areas shown in red in d02, d03, and d04. Their abbreviations are shown in parentheses.**






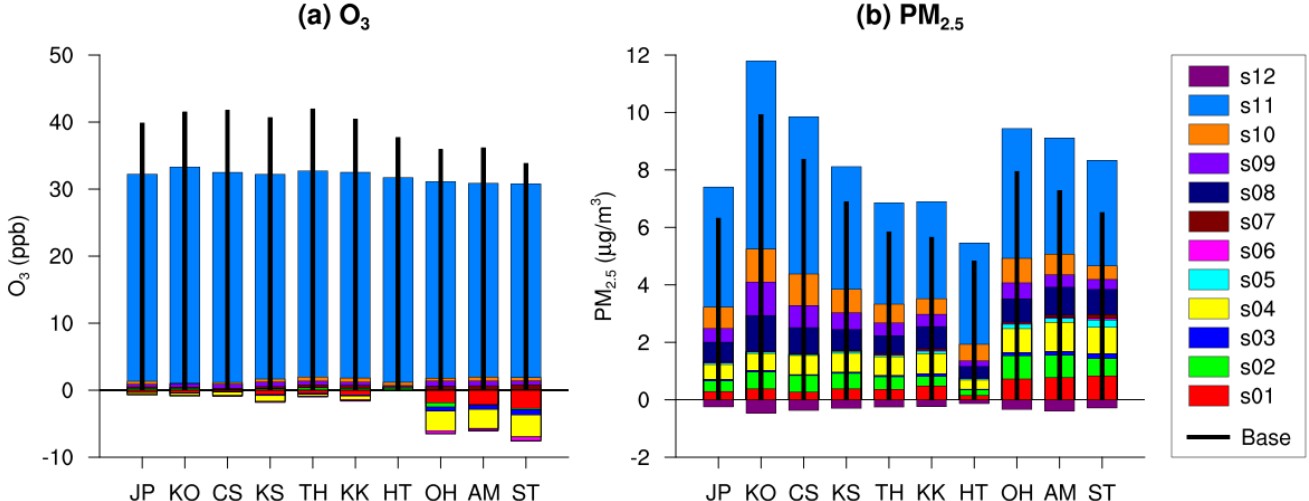

**Figure 2. Source sensitivities to the annual mean ozone and PM2.5 concentrations derived by BFM in the regions. Thick black lines represent the simulated concentrations.**



**Figure 3. Source sensitivities to the monthly mean ozone and PM2.5 concentrations derived by BFM in entire Japan (JP) and ST. Thick black lines represent the simulated concentrations.**




**Figure 4. Sensitivities per total annual amounts of precursor emissions from domestic anthropogenic sources (s01–s08) in d02 to the annual mean ambient concentrations of corresponding PM$_{2.5}$ components in entire Japan. All of them are normalized by the EC value for s01.**






**Figure 5. Sensitivities of all source groups over OH, AM, and ST evaluated in d02, d03, and d04 for the two target weeks in the four seasons.**





**Figure 6. Apportionments derived by ISAM and sensitivities derived by BFM and HDDM of all source groups to the simulated ozone and PM$_{2.5}$ concentrations in JP and ST for the two target weeks in the four seasons.**






**Figure 7: (a) Sensitivities of the source groups located within d02 (s01–s10) derived by BFM (left), HDDM-20 (middle), and HDDM-100 (right) to the daily NO$_3^-$ concentrations and (b) daily nonlinearity index and available NH$_3$ ratios for the two target weeks in winter in ST. Nonlinearity indices for first-order sensitivity coefficients less than 0.001 μg/m$^3$ are not shown as they are likely to be affected by numerical noise.**






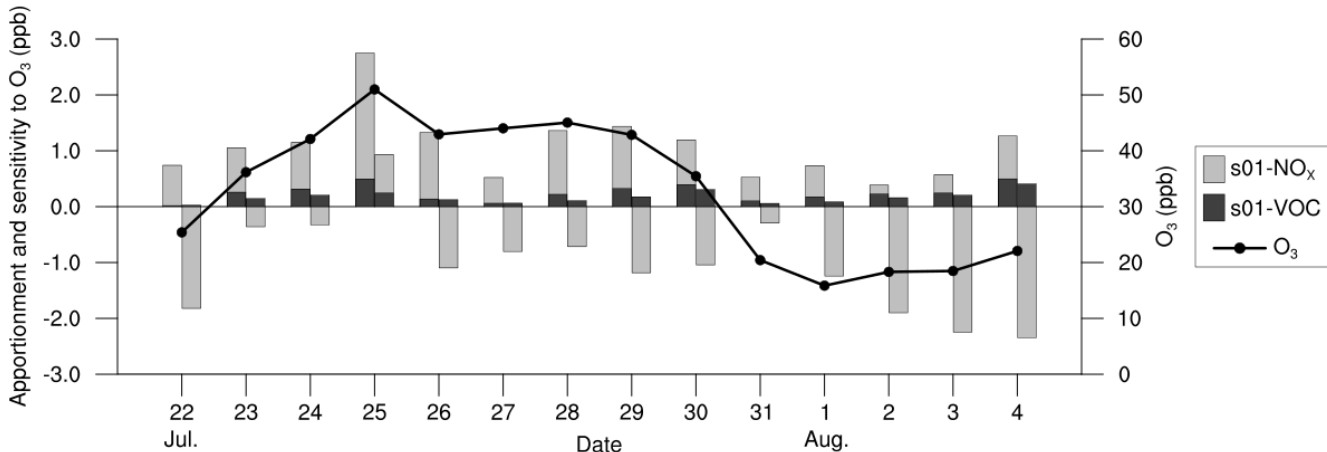

**Figure 8.** Apportionments (left) and sensitivities (right) for the NOₓ and VOC emissions of s01 derived by ISAM and BFM to daily ozone concentrations (shown by a line with markers) for the two target weeks during the summer in ST.

