# Peer review of "Comprehensive analyses of source sensitivities and apportionments of PM2.5 and ozone over Japan via multiple numerical techniques"

_Atmospheric Chemistry and Physics, 2020_

## Referee Comment (RC1) · Anonymous Referee #2 · 13 May 2020

The paper by Chatani et al. is based on source sensitivities and apportionments of O3 and PM2.5 over Japan by comparing 3 numerical techniques, 4 grids, 12 source groups. The paper is well organised and written, and the overall discussion is properly articulated. Figures are clear and they are all them necessary. I have only two minor comments for the authors: Minor comments -Line 178-179. According to the simulations and statement, "The PM2.5 concentrations were underestimated in all regions. The statistics tended to be worse in eastern Japan as opposed to western Japan." If the problem with the simulation has a clear geographical gradient (W-E), and after reading the discussion is mainly due to OC and nitrate, there is a probability of missing sources/atmospheric processes from local origin. Western Japanese sites are affected

by long-range transport aerosols from other Asian countries, but Eastern sites are also affected by Japanese sources (considering a prevalent western to eastern air flow).

-Line 381-384. The authors say "If ozone transported from outside Japan is not as reduced in future, efforts to reduce precursor emissions in Japan will not effectively contribute to the reduction in the concentrations of secondary PM2.5 components because OH that originated in ozone transported from outside Japan affects their formation", which is an interesting statement. But it is hard to figure out which sources are releasing PM2.5 precursors (for example NOx, SOx or VOCs) but not releasing O3 precursors. All combustion sources are strong VOC emitters, and efforts are made/have been made to abate NOx and SOx. Of course that the efforts in reducing emissions in Japan will not counteract the arrival of steady emissions from outside, but the reduction in precursor emissions in Japan will led to a lesser formation of secondary aerosols (although not in the same proportion as the applied reduction) and will contribute to the reduction of the continental O3 background.
* * *

---

## Referee Comment (RC2) · Anonymous Referee #3 · 18 May 2020

General comment: technically sound but conclusions unclear and disappointing, possibly overstated

The authors have performed air quality simulations using the CMAQ model over various nested domains including Japan or parts of that country. They show in a convincing way that their simulations are realistic and have resonable (even good) performance They study various methods to study the impacts of different types of sources in terms of concentrations of PM2.5 and ozone, including "Brute force method" (i.e. sensitivity simulations), ISAM and HDDM. As far as I can tell, all the methods implemented by the authors are technically sound and, at least in terms of modelled concentrations, they

are easily on par with the State of the art.

However, even though this study has obviously involved a big amount of work, its point is not clear to me. In the abstract, the authors state that "This study demonstrated that a combination of sensitivities and apportionments derived by the BFM, HDDM, and ISAM can provide critical information to identify key emission sources and processes in the atmosphere, which are vital for the development of effective strategies for improved air quality". A similar statement appears in the conclusionÂă: "This study demonstrated that a combination of sensitivities and apportionments derived by the BFM, HDDM, and ISAM can provide critical information to identify key emission sources and processes in the atmosphere, which are vital for the development of effective strategies for improved air quality, using consistent model configurations and inputs.". However, in-between I (and only "I" because that feeling is very possibly due to the fact that I am not so familiar with the issues the authors discuss) felt overwhelmed by a mass of plots and figures quite often lacking physico-chemical interpretation.

In summary, I have failed to understand which of the actual information unveiled by the authors was "critical" or even "vital" for policy design. On the contrary, I have the feeling that the methods they deploy are advanced but the actual results that they show are often disappointing when compared to the weaponry that they have used. For example, in the conclusion, the authors state that "Domestic sources had certain sensitivities to PM 2.5 , but significantly smaller or even negative sensitivities to ozone due to titration and nonlinear responses against precursor emissions.", which is hardly a surprise, it is discussed in all the good atmospheric composition textbooks that ozone concentrations are having a twofold sensitivity to emissions depending on the chemical regime. Here the authors' methodology seems to lead the reader to conclusions that are already very well-known.

I think the authors have realized good simulations of air quality over their areas of interest, convincingly shown that point, they have deployed methods they claim to be extremely useful in terms of understanding the rôle of different source areas and activity

sectors in air pollution in Japan, but in my opinion they fail to make that second point, leading to disappointing conclusions.

Title:

I have a hard time understanding the title, "Comprehensive analyses of source sensitivities to and apportionments of PM 2.5 and ozone over Japan via multiple numerical techniques". Even though it might be due to my partial knowledge of the jargon in this particular field, I have the feeling that, in the title and the rest of the text (e.g. l. 55, l. 74 and following, etc.). It seems that in the author's vocabulary they adress the sensitivity of the NOx emissions to ozone concentrations (this is just an example) while the ordinary way of thinking is more to assess the sensitivity of ozone concentrations to NOx emissions.

Major comments:

l. 461-464: "While PM 2.5 concentrations and their absolute sensitivities of all the sources were lower than those calculated by previous studies for past years due to emission reductions, the relative contributions of transport from outside Japan to the total sensitivities were even larger, suggesting that emissions in Japan have been reduced similar to surrounding countries, including China." I think the sensitivities and apportionment calculated by the authors do not depend on the actual emissions by Japan and China but on the emission hypotheses and inventories that have been chosen by the authors. I do not think the authors can draw any conclusion from their study regarding the emission reduction paths followed by Japan or China. I think the logical path leading to this result is circular ă: the authors make certain choices regarding emissions in Japan and China, they observe that the results they obtain are consistent with the hypothesis they made, but in my opinion this is no proof that their initial hypothesis is correct.

Minor comments, typos :

p. 1, l. 16-17: "While domestic sources had certain source apportionments to ozone concentrations, transport from outside Japan dominated the source sensitivities." If possible, many sentences of this kind should be formulated in a more intuitive way, eǍ.g., Âń"while domestic sources can contribute to a certain extent to simulated ozone concentrations, transport from outside Japan can be considered as the main overall driver of ozone concentrations in JapanÂǎÂż (this is only my interpretation of course, just as an example on how the authors should make their conclusions more accessible to readers in the field but not specialized). At all places where this is possible, the authors should formulate their statements and partial conclusions in more physical terms.

p. 1, l. 22: "that that"

l. 96: "Following" seems useless.

---

## Referee Comment (RC3) · Anonymous Referee #1 · 20 May 2020

This manuscript presents an in depth discussion of the differences between sensitivity analysis and source apportionment methods in terms of non-linear effects concerning transport and local emissions. The presented data is new and relevant to future ozone and PM2.5 control in Japan. However, I cannot recommend its publication in its current form. I suggest the following revisions before reconsideration. Major Comments: 1) An English mistake that makes the manuscript very difficult to read must be corrected. The authors refer to "sensitivity of emissions to the pollutant (ozone or PM)". This mistake starts in the title and continues throughout the main text and the supplemental material. It should be corrected as "sensitivity of a pollutant to emissions" meaning that the pollutant is the dependent variable which respond to changes in emissions as independent

variables. Similarly, there is another mistake in the use of the term "source apportionment". The existing literature refers to "source apportionment of a pollutant". The title is correct in this sense but in the text the authors refer to "source apportionment to the pollutant". This should be corrected as "source apportionment of a pollutant (ozone or PM) to the emissions (e.g., on-road vehicle emissions or NOx emissions from on-road vehicles). 2) It is not clear how HDDM-100 differs from HDDM-20. They both seem to be using the same sensitivity coefficients, i.e., slopes and curvatures at unperturbed level of emissions. If these coefficients were being calculated in different simulations with different levels of emissions then it might have been interesting to compare them. But the description in lines 330-335 suggests that they are the same thing. Similarly, it might be interesting to compare the results of BFM with 100% reduction for some of the most nonlinear pollutant-emission relations. At that level of reduction BFM results might be more similar to source apportionment. 3) The issue of how model performance might affect sensitivities and source apportionments in this study is an important one. An elaborate discussion would be very helpful instead of just a generic statement that it is important. For example, given the poor performance in nitrate, which source apportionments and sensitivities are more uncertain? How does the poor performance in nitrate affect the nonlinear sensitivities to NOx and NH3 emissions? 4) The conclusions are somewhat generic; they could be written in a way that praise the findings of this study. See the minor comments below for places in the abstract and conclusions where more specific information might give this study the credit that it deserves. Minor Comments: 1) The last statement of the abstract (lines 24-26) is very generic; it should be replaced with a statement of specifically what was found in this study. 2) We don't find out about the horizontal grid resolutions until Section 3.5. This information could be given in Section 2.1. 3) Did you report the HDDM convergence problems to the CMAQ modeling community? Others who experienced similar issues may be able to recommend solutions. 4) Line 177: Actually, I believe your model performance meets some of the goals in Emery et al (2017). You may want to distinguish between criteria and goals. 5) Line 195: Add "following sulfate" after "OC is the second major component of PM2.5" 6) Lines 196-197: "Less overestimates dots. . ." Consider deleting this sentence. 7) Lines 221-224: Please explain how the "chlorine loss" works in more detail and consider moving this discussion to the previous paragraph since the negative sensitivities to sea salt are first seen in Figure 2. 8) Lines 226-249: This discussion is difficult to follow. Perhaps you should use past tense for the previous studies and present tense for the current study. Also, state the two possible reasons upfront: 1) Japanese emissions are underestimated and 2) Foreign countries other than China are included (if I understand them correctly). I also recommend an explanation of the normalization mentioned in Table S3. 9) Line 283: Do we know what the background concentration levels are? 10) Figure 4: What is the rationale of selecting s01 EC for normalization? 11) Line 328: Is there a caveat of assuming that "OTHR" in ISAM is SOA. 12) Line 381: Replace "an oxidative capacity" with "the oxidative capacity" 13) Lines 454-455: Replace "can provide the" with "provides" 14) Line 462: "similar" or "more"? 15) Lines 479-480: Give an example for each of the direct and indirect influences that could not be distinguished only by the sensitivities. 16) Lines: 482-483: Give examples of how model performance may skew specific source sensitivities and apportionments in this study. 17) Lines 489-490: Consider deleting the sentence that begins with "In reality"

---

## Author Comment (AC1) · 22 Jun 2020

Dear Referee #1:

[Referee #1] This manuscript presents an in depth discussion of the differences between sensitivity analysis and source apportionment methods in terms of non-linear effects concerning transport and local emissions. The presented data is new and relevant to future ozone and PM2.5 control in Japan. However, I cannot recommend its publication in its current form. I suggest the following revisions before reconsideration.

[Reply] Thank you for valuable comments on our manuscript. I will revise the

manuscript based on your comments.

[Referee #1] Major Comments: 1) An English mistake that makes the manuscript very difficult to read must be corrected. The authors refer to "sensitivity of emissions to the pollutant (ozone or PM)". This mistake starts in the title and continues throughout the main text and the supplemental material. It should be corrected as "sensitivity of a pollutant to emissions" meaning that the pollutant is the dependent variable which respond to changes in emissions as independent variables. Similarly, there is another mistake in the use of the term "source apportionment". The existing literature refers to "source apportionment of a pollutant". The title is correct in this sense but in the text the authors refer to "source apportionment to the pollutant". This should be corrected as "source apportionment of a pollutant (ozone or PM) to the emissions (e.g., on-road vehicle emissions or NOx emissions from on-road vehicles).

[Reply] I am so sorry for this English mistake for the important words of this study. Another referee raised the same issue. I fully agree that "sensitivity of pollutant concentrations to emissions" and "apportionment of pollutant concentrations to emissions" are correct expressions.

The title will be changed as follows.

"Comprehensive analyses of source sensitivities and apportionments of PM2.5 and ozone over Japan via multiple numerical techniques"

In addition, I will check the main text, tables, figures, and supplemental material, to correct all the relevant parts. A grammar check will be also done again. Please check in the revised manuscript.

[Referee #1] 2) It is not clear how HDDM-100 differs from HDDM-20. They both seem to be using the same sensitivity coefficients, i.e., slopes and curvatures at unperturbed level of emissions. If these coefficients were being calculated in different simulations with different levels of emissions then it might have been interesting to compare them.

But the description in lines 330-335 suggests that they are the same thing. Similarly, it might be interesting to compare the results of BFM with 100% reduction for some of the most nonlinear pollutant-emission relations. At that level of reduction BFM results might be more similar to source apportionment.

[Reply] It is true that HDDM-100 and HDDM-20 are calculated using the same sensitivity coefficients, but their values could be different. The explanations in the lines 330-335 will be revised as follows. I hope they make discussions clearer.

"The HDDM-20 corresponds to the value calculated by applying $\Delta\varepsilon$=-0.2 and multiplication by 5. If a sensitivity is represented by a second-order polynomial function, HDDM-20 is equivalent to the value obtained by BFM. However, influences of the second-order term for a perturbation beyond 20% are not reflected in HDDM-20 because the value at a 20% perturbation is just linearly extrapolated. They are reflected in the HDDM-100, which corresponds to the value calculated by applying $\Delta\varepsilon$=-1.0. Differences between BFM and HDDM-20 correspond to the deviations of sensitivities from second-order functions, and differences between HDDM-20 and HDDM-100 correspond to the influences of the second-order term for a perturbation beyond 20%"

The sentence in the lines 411-412 will be revised as follows to make consistent with the explanations above.

"Differences should be recognized as difficulties in representing sensitivities only with first- and second-order sensitivity coefficients derived by HDDM."

I fully agree that it might be interesting to compare the results of BFM with 100% reduction for some of the most nonlinear pollutant-emission relations. Therefore, additional simulations were conducted with 100% reduction for s04 and s08 for the discussions in the Section 3.6.2, and for s01-NOX and s01-VOC for the discussions in the Section 3.6.3. Interesting results were obtained. I confirmed that sensitivities become closer to apportionments in some conditions.

[Figure]

The following descriptions will be added to the end of the Section 3.6.2 based on the results for s04 and s08 with 100% reduction.

"In addition to BFM with 20% perturbation (denoted as BFM-20), additional simulations were conducted to derive sensitivities by BFM with 100% perturbation (denoted as BFM-100) for s04, which emits NOX and no NH3, and s08, which emit NH3 and no NOX. Figure S7 in the Supplementary Material shows the sensitivities derived by BFM-20, BFM-100, HDDM-20, and HDDM-100, and apportionments derived by ISAM of the daily NO3- and NH4+ concentrations to s04 and s08 for the two target weeks in winter in ST. The sensitivities derived by BFM-100 are higher than those derived by BFM-20 due to nonlinear responses. Similar features are evident in the sensitivities derived by HDDM-100 and HDDM-20, implying that HDDM is capable to represent directions of nonlinear responses beyond 20% perturbation. It is notable that the sensitivities derived by BFM with a larger perturbation become closer to the apportionments for NO3- to s04, and NH4+ to s08. However, there are still deviations among them caused by indirect influences of factors including other sectors, complex photochemical reactions, and gas-aerosol partitioning. Moreover, the apportionments of NO3- to s08 and of NH4+ to s04 never appear while NO3- and NH4+ concentrations are nonlinearly sensitive to s08 and s04, respectively."

The third paragraph of the Section 3.6.3 will be divided into two and the latter one will be revised as follows based on the results for s01-VOC and s01-NOX with 100% reduction. Figure 8 (attached as Fig. 1 in this reply) will be replaced accordingly.

"We note that the sensitivities to VOC emissions derived by BFM-20 and BFM-100 are almost identical. That means ozone formation from VOCs is linear to emissions. The sensitivities of NOX emissions derived by BFM-20 and BFM-100 are also almost identical when they are negative. That means titration of ozone by NOX is also linear to emissions. In contrast, the sensitivities to NOX emissions derived by BFM-100 are higher than those derived by BFM-20 when they are positive. That means ozone formation from NOX is nonlinear to emissions. Cohan et al. (2005) also reported that the

sensitivities of ozone concentrations are lower when perturbations of precursor emissions are smaller because other remaining precursors are more likely to contribute to ozone formation instead. This may also be the reason why the sums of the sensitivities to all the sources are lower than the simulated ozone concentrations in spring and summer, as shown in Figs. 2, 3, and 5. While the sensitivities derived by BFM-100 become closer to the apportionments, the apportionments are still higher than the sensitivities as discussed for NO3- and NH4+ in the Section 3.6.2. That implies effects on concentrations of ozone, NO3-, and NH4+ may be less than those inferred by BFM-100 and ISAM when reductions of emissions of NOX and NH3 are small."

[Referee #1] 3) The issue of how model performance might affect sensitivities and source apportionments in this study is an important one. An elaborate discussion would be very helpful instead of just a generic statement that it is important. For example, given the poor performance in nitrate, which source apportionments and sensitivities are more uncertain? How does the poor performance in nitrate affect the nonlinear sensitivities to NOx and NH3 emissions?

[Reply] While it is a difficult question to answer because reasons of poor model performance have not been clarified, I will add an analysis to answer to this important question. Please see the reply to the minor comment below.

[Referee #1] 4) The conclusions are somewhat generic; they could be written in a way that praise the findings of this study. See the minor comments below for places in the abstract and conclusions where more specific information might give this study the credit that it deserves.

[Reply] I will revise the conclusions to praise the findings of this study more clearly and to add findings of additional simulations. Please check them in the revised manuscript.

[Referee #1] Minor Comments: 1) The last statement of the abstract (lines 24-26) is very generic; it should be replaced with a statement of specifically what was found in this study.

[Figure]

[Reply] They will be replaced by the following sentences including relationships between sensitivities and apportionments described in the reply above.

"While the sensitivities become closer to the apportionments when perturbations in emissions are larger in highly nonlinear relationships including those between NH3 emissions and NH4+ concentrations, NOX emissions and NO3- concentrations, and NOX emissions and ozone concentrations, the sensitivities did not reach the apportionments due to various indirect influences including other sectors, complex photochemical reactions and gas-aerosol partitioning. It is essential to consider nonlinear influences to derive strategies for effectively supressing concentrations of secondary pollutants."

[Referee #1] 2) We don't find out about the horizontal grid resolutions until Section 3.5. This information could be given in Section 2.1.

[Reply] A description on the horizontal grid resolutions will be inserted to the line 90 as follows.

"Horizontal resolutions of d01, d02, d03, and d04 are 45 x 45 km, 15 x 15 km, 5 x 5 km, and 5 x 5 km, respectively."

[Referee #1] 3) Did you report the HDDM convergence problems to the CMAQ modeling community? Others who experienced similar issues may be able to recommend solutions.

[Reply] While we did not report to the CMAQ modeling community (e.g. CMAS forum), one of members in the CMAQ developing team agreed that there are still convergence problems in HDDM embedded in CMAQ. The problem could be avoided by altering some model configurations, but that could not be done because consistencies among BFM, HDDM, and ISAM are important in this study.

[Referee #1] 4) Line 177: Actually, I believe your model performance meets some of the goals in Emery et al (2017). You may want to distinguish between criteria and goals.

[Reply] Thank you for your suggestion. I confirmed that our model performance meets the goals for limited species and in limited regions. However, I would avoid making discussions complicated by distinguishing goals and criteria.

[Referee #1] 5) Line 195: Add "following sulfate" after "OC is the second major component of PM2.5"

[Reply] It will be added as suggested as follows to the line 195.

"As OC is the second major component of PM2.5 following $SO_4^{2-}$"

[Referee #1] 6) Lines 196-197: "Less overestimates dots: : :" Consider deleting this sentence.

[Reply] This sentence will be deleted as suggested.

[Referee #1] 7) Lines 221-224: Please explain how the "chlorine loss" works in more detail and consider moving this discussion to the previous paragraph since the negative sensitivities to sea salt are first seen in Figure 2.

[Reply] The following sentence will be added to the previous paragraph to just show the fact of the negative sensitivities to sea salts.

"The sensitivity of PM2.5 to s12 (sea salt) is negative."

Explanations in the lines 221-224 will be revised as follows. I hope they make discussions clearer.

"The sensitivities of $NO_3^-$ and $NH_4^+$ to s12 (sea salt) are negative. $Cl^-$ originated from sea salts and mostly involved in coarse particles tend to be replaced by $NO_3^-$ due to so-called chlorine loss caused by gas-aerosol partitioning (Pio and Lopes, 1998; Chen et al., 2016). Therefore, if sea salts are present, more $HNO_3$ gases are partitioned to coarse particles. That provides capacities for $NO_3^-$ and associated $NH_4^+$ involved in PM2.5 to evaporate to the gas phase, resulting in negative sensitivities of PM2.5 including $NO_3^-$ and $NH_4^+$ to sea salts."

[Referee #1] 8) Lines 226-249: This discussion is difficult to follow. Perhaps you should use past tense for the previous studies and present tense for the current study. Also, state the two possible reasons upfront: 1) Japanese emissions are underestimated and 2) Foreign countries other than China are included (if I understand them correctly). I also recommend an explanation of the normalization mentioned in Table S3.

[Reply] I am sorry for these difficult sentences. I will correct all the tense so that past tense is used for the previous studies and present tense is used for the current study.

There are two reasons: (1) reduction of emissions in Japan, and (2) other factors than emissions in China. The paragraph in the lines 227-250 will be divided into three to make easier to follow. The latter two paragraph, originally in the lines 237 and 248, will start with the following sentences, respectively.

"One of possible reasons for these elevated contributions is reduction of emissions in Japan" "Another reason of the higher contributions of foreign sources owes to other factors than emissions in China."

The explanation of the normalization will be added to the beginning of this paragraph as follows.

"Table S3 in Supplementary Material lists the ratios of the source sensitivities of the annual mean ozone and PM2.5 concentrations simulated in the regions, which were compared with previous studies. While sums of the ratios of the sensitivities to all the source groups are not 100% due to nonlinearities, they were often normalized to 100% in previous studies. Therefore, the ratios normalized to make their sums equal to 100% are also shown in Table S3."

[Referee #1] 9) Line 283: Do we know what the background concentration levels are?

[Reply] It may be inappropriate to mention background concentrations because they are unknown in this study. Therefore, this and the preceding sentences will be deleted.

[Referee #1] 10) Figure 4: What is the rationale of selecting s01 EC for normalization?
[Reply] There is no rationale. Anything can be used for normalization because Figure 4 just shows relative relationships among sensitivities. s01 EC was selected just because they are inert and emitted only in the bottom layer. Such an explanation will be inserted in the line 295 as follows.

"All the values shown in Figure 4 were normalized by the EC value for s01, which is inert and emitted only in the bottom layer."

[Referee #1] 11) Line 328: Is there a caveat of assuming that "OTHR" in ISAM is SOA.

[Reply] It would be ideal that ISAM can calculate apportionments of SOA. However, it is impossible for ISAM embedded in CMAQ version 5.0.2. Descriptions around the line 328 will be revised as follows.

"The simulated SOA concentrations were characterised as apportionments of "OTHR" in ISAM in this study because apportionments of SOA concentrations were not calculated by ISAM embedded in CMAQ version 5.0.2"

[Referee #1] 12) Line 381: Replace "an oxidative capacity" with "the oxidative capacity"

[Reply] It will be replaced as suggested.

[Referee #1] 13) Lines 454-455: Replace "can provide the" with "provides"

[Reply] It will be replaced as suggested.

[Referee #1] 14) Line 462: "similar" or "more"?

[Reply] The sentence in the lines 461-464 will be revised as follows.

"While PM2.5 concentrations were lower than those simulated by previous studies for past years due to emission reductions, the relative contributions of transport from outside Japan to the total sensitivities were even larger, suggesting that emissions in Japan have been similarly reduced to surrounding countries including China."

[Referee #1] 15) Lines 479-480: Give an example for each of the direct and indirect

influences that could not be distinguished only by the sensitivities.

[Reply] The following example will be inserted in the line 481. This paragraph will be merged to the precedent one because both contain similar descriptions.

"For example, the sensitivities of SO42- and NO3- to the transport from outside Japan encompassed at least two undistinguished influencing factors, including the direct transport of SO42- and NO3-, which were evaluated by their corresponding apportionments, and oxidation of SO2 and NOX emitted from domestic sources by OH originating in ozone transported from outside."

[Referee #1] 16) Lines: 482-483: Give examples of how model performance may skew specific source sensitivities and apportionments in this study.

[Reply] Figure S10 will be added to show sensitivities of PM2.5 uniformly scaled by the ratios of observed and simulated concentrations of PM2.5 components. Discussions on this figure will be inserted in the line 484 as follows.

"Figure s10 in the Supplementary Material shows source sensitivities of the annual mean PM2.5 concentrations derived by BFM in the regions. The values shown in (b) were uniformly scaled by the ratios of observed and simulated concentrations of PM2.5 components shown in Table S2. The scaled sensitivities of PM2.5 to the transport from outside Japan are higher by 1.0-2.2 $\mu$g/m3 (15-40%) due to their high contributions to underestimated POA and SOA. The scaled sensitivities of PM2.5 to other sources are different by 0-0.5 $\mu$g/m3. This is the case that deviations between observed and simulated PM2.5 concentrations can be proportionally explained by the source sensitivities. Uncertainties could be higher if specific sources cause poor model performance. In particular, this study revealed NH4+ and NO3- concentrations are nonlinearly sensitive to NH3 and NOX emissions. Uncertainties in NH3 and NOX emission sources could largely influence source sensitivities as well as model performance of NH4+ and NO3- concentrations. More studies are necessary to obtain more confidence in source sensitivities and apportionments as well as model performance."
* * *
Interactive
comment

[Referee #1] 17) Lines 489-490: Consider deleting the sentence that begins with "In reality"

[Reply] This sentence will be replaced by the following one.

"However, model configurations and inputs may not be necessarily consistent."

Please also note the supplement to this comment:
https://www.atmos-chem-phys-discuss.net/acp-2020-236/acp-2020-236-AC1-supplement.pdf
* * *
[Figure]

**Fig. 1.**

---

## Author Comment (AC2) · 22 Jun 2020

Dear Referee #2:

[Referee #2] The paper by Chatani et al. is based on source sensitivities and apportionments of O3 and PM2.5 over Japan by comparing 3 numerical techniques, 4 grids, 12 source groups. The paper is well organised and written, and the overall discussion is properly articulated. Figures are clear and they are all them necessary.

[Reply] Thank you for valuable comments on our manuscript. I will revise the manuscript based on your comments.

[Figure]

[Referee #2] I have only two minor comments for the authors. Line 178-179. According to the simulations and statement, "The PM2.5 concentrations were underestimated in all regions. The statistics tended to be worse in eastern Japan as opposed to western Japan." If the problem with the simulation has a clear geographical gradient (W-E), and after reading the discussion is mainly due to OC and nitrate, there is a probability of missing sources/atmospheric processes from local origin. Western Japanese sites are affected by long-range transport aerosols from other Asian countries, but Eastern sites are also affected by Japanese sources (considering a prevalent western to eastern air flow).

[Reply] I fully agree your comment. Actually, this issue has been already discussed in the lines 245-248, but I will revise it to make this issue clearer.

The following sentence will be inserted in the line 181.

"A possible reason will be discussed in the Section 3.2."

The description in the lines 245-248 will be revised as follows.

"However, we can also state that the underestimations of the PM2.5 concentrations are larger in eastern than western Japan as described in section 3.1. Influences of domestic sources should be accumulated more in eastern than western Japan because a prevalent air flow over Japan is westerly. Therefore, worse model performance in eastern Japan imply underestimation of domestic emissions. Reductions of domestic emissions from fiscal years 2005 to 2015 may be excessively estimated."

Referee #2 Line 381-384. The authors say "If ozone transported from outside Japan is not as reduced in future, efforts to reduce precursor emissions in Japan will not effectively contribute to the reduction in the concentrations of secondary PM2.5 components because OH that originated in ozone transported from outside Japan affects their formation", which is an interesting statement. But it is hard to figure out which sources are releasing PM2.5 precursors (for example NOx, SOx or VOCs) but not releasing O3 pre-

cursors. All combustion sources are strong VOC emitters, and efforts are made/have been made to abate NOx and SOx. Of course that the efforts in reducing emissions in Japan will not counteract the arrival of steady emissions from outside, but the reduction in precursor emissions in Japan will led to a lesser formation of secondary aerosols (although not in the same proportion as the applied reduction) and will contribute to the reduction of the continental O3 background.

[Reply] I agree your comments. Source releasing PM2.5 precursors emit O3 precursors. Discussions were too generalized. They should focus on SO42- and NO3- as target species and SO2 and NOX as precursors. Corresponding expressions in this paragraph in the lines 377-385 will be revised as follows.

"Section 3.2 discussed higher relative contributions than previous studies and less contrasts between western and eastern Japan for the sensitivities of PM2.5 to s11 obtained in this study. Oxidation of SO2 and NOX emitted from domestic sources by OH that originated in ozone transported from outside Japan is another factor that causes higher sensitivities of s11. The entirety of Japan is equally affected by ozone transported from outside Japan, as shown in Fig. 2(a), due to its long lifetime in the atmosphere, resulting in less contrasts in the sensitivities of s11 between western and eastern Japan, while the sensitivities of domestic emissions are small. Ozone governs the oxidative capacity of the atmosphere (Prinn, 2003). If ozone transported from outside Japan is not as reduced in future, efforts to reduce SO2 and NOX emissions in Japan will not effectively contribute to the reduction in the concentrations of SO42- and NO3- because OH that originated in ozone transported from outside Japan affects their formation."

However, the sensitivities of ozone to domestic emissions are small. In addition, influences of emissions in Japan to background ozone are marginal. I think influences of emissions of ozone precursors in Japan on oxidation of SO2 and NO2 are limited. The following sentence will be inserted in the line 382.

"while the sensitivities of ozone to domestic emissions are small."

---

## Author Response (AR1)

Dear Referee #1:

*[Referee #1]*

*This manuscript presents an in depth discussion of the differences between sensitivity analysis and source apportionment methods in terms of non-linear effects concerning transport and local emissions. The presented data is new and relevant to future ozone and PM2.5 control in Japan. However, I cannot recommend its publication in its current form. I suggest the following revisions before reconsideration.*

[Reply]

Thank you for valuable comments on our manuscript. I have revised the manuscript based on your comments.

*[Referee #1]*

*Major Comments:*

*1) An English mistake that makes the manuscript very difficult to read must be corrected. The authors refer to "sensitivity of emissions to the pollutant (ozone or PM)". This mistake starts in the title and continues throughout the main text and the supplemental material. It should be corrected as "sensitivity of a pollutant to emissions" meaning that the pollutant is the dependent variable which respond to changes in emissions as independent variables. Similarly, there is another mistake in the use of the term "source apportionment". The existing literature refers to "source apportionment of a pollutant". The title is correct in this sense but in the text the authors refer to "source apportionment to the pollutant". This should be corrected as "source apportionment of a pollutant (ozone or PM) to the emissions (e.g., on-road vehicle emissions or NOx emissions from on-road vehicles).*

[Reply]

I am so sorry for this English mistake for the important words of this study. Another referee raised the same issue. I fully agree that "sensitivity of pollutant concentrations to emissions" and "apportionment of pollutant concentrations to emissions" are correct expressions.

The title has been changed as follows.

"Comprehensive analyses of source sensitivities and apportionments of $PM_{2.5}$ and ozone over Japan via multiple numerical techniques"

In addition, I have checked the main text, tables, figures, and supplemental material, to correct all the relevant parts. A grammar check has been also done again. Please check in the revised manuscript.

*[Referee #1]*

*2) It is not clear how HDDM-100 differs from HDDM-20. They both seem to be using the same sensitivity coefficients, i.e., slopes and curvatures at unperturbed level of emissions. If these coefficients were being calculated in different simulations with different levels of emissions then it might have been interesting to compare them. But the description in lines 330-335 suggests that they are the same thing. Similarly, it might be interesting to compare the results of BFM with 100% reduction for some of the most nonlinear pollutant-emission relations. At that level of reduction BFM results might be more similar to source apportionment.*

[Reply]

It is true that HDDM-100 and HDDM-20 are calculated using the same sensitivity coefficients, but their values could be different. The explanations in the lines 358-365 have been revised as follows. I hope they make discussions clearer.

"The HDDM-20 corresponds to the value calculated by applying $\Delta\varepsilon = -0.2$ and multiplication by 5. If a sensitivity is represented by a second-order polynomial function, HDDM-20 is equivalent to the value obtained by BFM. However, the influence of the second-order term for a perturbation beyond 20% is not reflected in HDDM-20 because the value at a 20% perturbation is just linearly extrapolated. They are reflected in the HDDM-100, which corresponds to the value calculated by applying $\Delta\varepsilon = -1.0$. Differences between BFM and HDDM-20 correspond to the deviations of sensitivities from second-order functions, and differences between HDDM-20 and HDDM-100 correspond to the influences of the second-order term for a perturbation beyond 20%"

The sentence in the lines 439-440 has been revised as follows to make consistent with the explanations above.

"Differences should be recognized as difficulties in representing sensitivities only with first- and second-order sensitivity coefficients derived by HDDM"

I fully agree that it might be interesting to compare the results of BFM with 100% reduction for some of the most nonlinear pollutant-emission relations. Therefore, additional simulations were conducted with 100% reduction for s04 and s08 for the discussions in the Section 3.6.2, and for s01-NO$_X$ and s01-VOC for the discussions in the Section 3.6.3. Interesting results were obtained. I confirmed that sensitivities become closer to apportionments in some conditions.

The following descriptions have been added to the end of the Section 3.6.2 based on the results for s04 and s08 with 100% reduction.

"In addition to BFM with 20% perturbation (denoted as BFM-20), additional simulations were conducted to derive sensitivities by BFM with 100% perturbation (denoted as BFM-100) for s04, which emits $NO_X$ but not $NH_3$, and s08, which emit $NH_3$ but not $NO_X$. Figure S7 in the Supplementary Material shows the sensitivities derived by BFM-20, BFM-100, HDDM-20, and HDDM-100, and apportionments derived by ISAM of the daily $NO_3^-$ and $NH_4^+$ concentrations to s04 and s08 for the two target weeks in winter in ST. The sensitivities derived by BFM-100 are higher than those derived by BFM-20 because of the nonlinear responses. Similar features are evident in the sensitivities derived by HDDM-100 and HDDM-20, implying that HDDM is capable of representing directions of nonlinear responses beyond 20% perturbation. It is notable that the sensitivities derived by BFM with a larger perturbation become closer to the apportionments for $NO_3^-$ to s04, and $NH_4^+$ to s08. However, there are still deviations among them caused by indirect influences of factors including other sectors, complex photochemical reactions, and gas-aerosol partitioning. Moreover, $NO_3^-$ and $NH_4^+$ concentrations are never apportioned but nonlinearly sensitive to s08 and s04, respectively."

The third paragraph of the Section 3.6.3 has been divided into two and the latter one has been revised as follows based on the results for s01-VOC and s01-$NO_X$ with 100% reduction. Figure 8 (attached as Fig. 1 in this reply) has been replaced accordingly.

"We note that the sensitivities to VOC emissions derived by BFM-20 and BFM-100 are almost identical. That means ozone formation from VOCs is linearly related to emissions. The sensitivities of $NO_X$ emissions derived by BFM-20 and BFM-100 are also almost identical when they are negative. That means titration of ozone by $NO_X$ is also linearly related to emissions. In contrast, the sensitivities to $NO_X$ emissions derived by BFM-100 are higher than those derived by BFM-20 when they are positive. That means ozone formation from $NO_X$ is nonlinearly related to emissions. Cohan et al. (2005) also reported that the sensitivities of ozone concentrations are lower when perturbations of precursor emissions are smaller because other remaining precursors are more likely to contribute to ozone formation instead. This may also be the reason why the sums of the sensitivities to all the sources are lower than the simulated ozone concentrations in spring and summer (Figs. 2, 3, and 5). While the sensitivities derived by BFM-100 become closer to the apportionments, the apportionments are still higher than the sensitivities as discussed for $NO_3^-$ and $NH_4^+$ in section 3.6.2. That implies effects on concentrations of ozone, $NO_3^-$, and $NH_4^+$ may be less than those inferred by BFM-100 and ISAM when reductions of emissions of $NO_X$ and $NH_3$ are small."

*[Referee #1]*

*3) The issue of how model performance might affect sensitivities and source apportionments in this study is an important one. An elaborate discussion would be very helpful instead of just a generic statement that it is important. For example, given the poor performance in nitrate, which source apportionments and sensitivities are more uncertain? How does the poor performance in nitrate affect the nonlinear sensitivities to NOx and NH3 emissions?*

[Reply]

While it is a difficult question to answer because reasons of poor model performance have not been clarified, I have added an analysis to answer to this important question. Please see the reply to the minor comment below.

*[Referee #1]*

*4) The conclusions are somewhat generic; they could be written in a way that praise the findings of this study. See the minor comments below for places in the abstract and conclusions where more specific information might give this study the credit that it deserves.*

[Reply]

I have revised the conclusions to praise the findings of this study more clearly and to add findings of additional simulations. Please check them in the revised manuscript.

*[Referee #1]*

*Minor Comments:*

*1) The last statement of the abstract (lines 24-26) is very generic; it should be replaced with a statement of specifically what was found in this study.*

[Reply]

They have been replaced by the following sentences in the lines 24-29 including relationships between sensitivities and apportionments described in the reply above.

"While the sensitivities become closer to the apportionments when perturbations in emissions are larger in highly nonlinear relationships – including those between $NH_3$ emissions and $NH_4^+$ concentrations, $NO_X$ emissions and $NO_3^-$ concentrations, and $NO_X$ emissions and ozone concentrations – the sensitivities did not reach the apportionments because there were various indirect influences including other sectors, complex photochemical reactions, and gas-aerosol partitioning. It is essential to consider nonlinear influences to derive strategies for effectively supressing

concentrations of secondary pollutants."

*[Referee #1]*
*2) We don't find out about the horizontal grid resolutions until Section 3.5. This information could be given in Section 2.1.*

[Reply]
A description on the horizontal grid resolutions has been inserted to the lines 100-101 as follows.

"Horizontal resolutions of d01, d02, d03, and d04 are 45 × 45 km, 15 × 15 km, 5 × 5 km, and 5 × 5 km, respectively."

*[Referee #1]*
*3) Did you report the HDDM convergence problems to the CMAQ modeling community? Others who experienced similar issues may be able to recommend solutions.*

[Reply]
While we did not report to the CMAQ modeling community (e.g. CMAS forum), one of members in the CMAQ developing team agreed that there are still convergence problems in HDDM embedded in CMAQ. The problem could be avoided by altering some model configurations, but that could not be done because consistencies among BFM, HDDM, and ISAM are important in this study.

*[Referee #1]*
*4) Line 177: Actually, I believe your model performance meets some of the goals in Emery et al (2017). You may want to distinguish between criteria and goals.*

[Reply]
Thank you for your suggestion. I confirmed that our model performance meets the goals for limited species and in limited regions. However, I avoided making discussions complicated by distinguishing goals and criteria.

*[Referee #1]*
*5) Line 195: Add "following sulfate" after "OC is the second major component of PM2.5"*

[Reply]
It has been added as suggested as follows to the lines 210-211.

"As OC is the second major component of $PM_{2.5}$ following $SO_4^{2-}$"

*[Referee #1]*
*6) Lines 196-197: "Less overestimates dots: : :" Consider deleting this sentence.*

[Reply]
This sentence has been deleted as suggested.

*[Referee #1]*
*7) Lines 221-224: Please explain how the "chlorine loss" works in more detail and consider moving this discussion to the previous paragraph since the negative sensitivities to sea salt are first seen in Figure 2.*

[Reply]
The following sentence has been added to the previous paragraph in the line 224 to just show the fact of the negative sensitivities to sea salts.

"The sensitivity of $PM_{2.5}$ to s12 (sea salt) is negative."

Explanations in the lines 238-242 have been revised as follows. I hope they make discussions clearer.

"The sensitivities of $NO_3^-$ and $NH_4^+$ to s12 (sea salt) are negative. $Cl^-$ originated from sea salts and mostly involved in coarse particles tend to be replaced by $NO_3^-$ because of the so-called chlorine loss caused by gas-aerosol partitioning (Pio and Lopes, 1998; Chen et al., 2016). Therefore, if sea salts are present, more $HNO_3$ gases are partitioned to coarse particles. That provides capacities for $NO_3^-$ and associated $NH_4^+$ involved in $PM_{2.5}$ to evaporate to the gas phase, resulting in negative sensitivities of $PM_{2.5}$ including $NO_3^-$ and $NH_4^+$ to sea salts."

*[Referee #1]*
*8) Lines 226-249: This discussion is difficult to follow. Perhaps you should use past tense for the previous studies and present tense for the current study. Also, state the two possible reasons upfront: 1) Japanese emissions are underestimated and 2) Foreign countries other than China are included (if I understand them correctly). I also recommend an explanation of the normalization mentioned in Table S3.*

[Reply]

I am sorry for these difficult sentences. I have corrected all the tense so that past tense is used for the previous studies and present tense is used for the current study.

There are two reasons: (1) reduction of emissions in Japan, and (2) other factors than emissions in China. The paragraph in the lines 257-273 has been divided into three to make easier to follow. The latter two paragraphs, originally in the lines 257 and 270, have started with the following sentences, respectively.

"One of possible reasons for these elevated contributions is reduction of emissions in Japan"
"Besides the changes in Chinese emissions, there are other reasons for the higher contributions from sources outside Japan."

The explanation of the normalization has been added to the beginning of this paragraph in the lines 244-247 as follows.

"Table S3 in Supplementary Material lists the ratios of the source sensitivities of the annual mean ozone and $PM_{2.5}$ concentrations simulated in the regions, which were compared with previous studies. While sums of the ratios of the sensitivities to all the source groups are not 100% because of the nonlinearities, they were often normalized to 100% in previous studies. Therefore, the ratios normalized to make their sums equal to 100% are also shown in Table S3."

*[Referee #1]*
*9) Line 283: Do we know what the background concentration levels are?*

[Reply]

It may be inappropriate to mention background concentrations because they are unknown in this study. Therefore, this and the preceding sentences have been be deleted.

*[Referee #1]*
*10) Figure 4: What is the rationale of selecting s01 EC for normalization?*

[Reply]

There is no rationale. Anything can be used for normalization because Figure 4 just shows relative relationships among sensitivities. s01 EC was selected just because they are inert and emitted only in the bottom layer. Such an explanation has been inserted in the lines 317-318 as follows.

"All the values shown in Fig. 4 were normalized by the EC value for s01, which is inert and emitted only in the bottom layer."

*[Referee #1]*
*11) Line 328: Is there a caveat of assuming that "OTHR" in ISAM is SOA.*

[Reply]
It would be ideal that ISAM can calculate apportionments of SOA. However, it is impossible for ISAM embedded in CMAQ version 5.0.2. Descriptions in the lines 352-354 have been revised as follows.

"The simulated SOA concentrations were characterized as apportionments of "OTHR" in ISAM in this study because apportionments of SOA concentrations were not calculated by ISAM embedded in CMAQ version 5.0.2"

*[Referee #1]*
*12) Line 381: Replace "an oxidative capacity" with "the oxidative capacity"*

[Reply]
It has been replaced as suggested.

*[Referee #1]*
*13) Lines 454-455: Replace "can provide the" with "provides"*

[Reply]
It has been replaced as suggested.

*[Referee #1]*
*14) Line 462: "similar" or "more"?*

[Reply]
The sentence in the lines 507-510 has been revised as follows.

"While $PM_{2.5}$ concentrations were lower than those simulated by previous studies for past years because of emission reductions, the relative contributions of transport from outside Japan to the total sensitivities were even larger, suggesting that emissions in Japan have been similarly reduced to

surrounding countries, including China."

*[Referee #1]*
*15) Lines 479-480: Give an example for each of the direct and indirect influences that could not be distinguished only by the sensitivities.*

[Reply]
The following example has been inserted in the lines 519-522. This paragraph has been merged to the precedent one because both contain similar descriptions.

"For example, the sensitivities of $SO_4^{2-}$ and $NO_3^-$ to the transport from outside Japan encompassed at least two undistinguished influencing factors, including the direct transport of $SO_4^{2-}$ and $NO_3^-$, which were evaluated by their corresponding apportionments, and oxidation of $SO_2$ and $NO_X$ emitted from domestic sources by OH originating in ozone transported from outside Japan."

*[Referee #1]*
*16) Lines: 482-483:*
*Give examples of how model performance may skew specific source sensitivities and apportionments in this study.*

[Reply]
Figure S10 has been added to show sensitivities of $PM_{2.5}$ uniformly scaled by the ratios of observed and simulated concentrations of $PM_{2.5}$ components. Discussions on this figure have been inserted in the lines 536-545 as follows.

"Figure s10 in the Supplementary Material shows source sensitivities of the annual mean $PM_{2.5}$ concentrations derived by BFM in the regions. The values shown in (b) were uniformly scaled by the ratios of observed and simulated concentrations of $PM_{2.5}$ components shown in Table S2. The scaled sensitivities of $PM_{2.5}$ to the transport from outside Japan are higher by 1.0–2.2 μg/m$^3$ (15–40%) because of their high contributions to underestimated POA and SOA. The scaled sensitivities of $PM_{2.5}$ to other sources are different by 0–0.5 μg/m$^3$. This case assumes that deviations between observed and simulated $PM_{2.5}$ concentrations can be proportionally explained by the source sensitivities. Uncertainties could be higher if specific sources cause poor model performance. In particular, this study revealed $NH_4^+$ and $NO_3^-$ concentrations are nonlinearly sensitive to $NH_3$ and $NO_X$ emissions. Uncertainties in $NH_3$ and $NO_X$ emission sources could largely influence source sensitivities as well as model performance of $NH_4^+$ and $NO_3^-$ concentrations. More studies are necessary to increase the

confidence in source sensitivities and apportionments as well as model performance."

*[Referee #1]*

*17) Lines 489-490: Consider deleting the sentence that begins with "In reality"*

[Reply]

This sentence in the lines 551-552 has been replaced by the following one.

"However, model configurations and inputs may not necessarily be consistent."

Dear Referee #2:

*[Referee #2]*

*The paper by Chatani et al. is based on source sensitivities and apportionments of O3 and PM2.5 over Japan by comparing 3 numerical techniques, 4 grids, 12 source groups. The paper is well organised and written, and the overall discussion is properly articulated. Figures are clear and they are all them necessary.*

[Reply]

Thank you for valuable comments on our manuscript. I have revised the manuscript based on your comments.

*[Referee #2]*

*I have only two minor comments for the authors.*

*Line 178-179.*

*According to the simulations and statement, "The PM2.5 concentrations were underestimated in all regions. The statistics tended to be worse in eastern Japan as opposed to western Japan." If the problem with the simulation has a clear geographical gradient (W-E), and after reading the discussion is mainly due to OC and nitrate, there is a probability of missing sources/atmospheric processes from local origin. Western Japanese sites are affected by long-range transport aerosols from other Asian countries, but Eastern sites are also affected by Japanese sources (considering a prevalent western to eastern air flow).*

[Reply]

I fully agree your comment. Actually, this issue has been already discussed in the lines 245-248, but I will revise it to make this issue clearer.

The following sentence will be inserted in the lines 194-195.

"A possible reason is discussed in section 3.2."

The description in the lines 265-269 has been revised as follows.

"However, we can also state that the underestimations of the $PM_{2.5}$ concentrations are larger in eastern than western Japan as described in section 3.1. Influences of domestic sources should be accumulated more in eastern than western Japan because the prevalent air flow over Japan is westerly. Therefore,

worse model performance in eastern Japan imply underestimation of domestic emissions. Reductions of domestic emissions from fiscal years 2005 to 2015 may be overestimated."

*[Referee #2]*

*Line 381-384.*

*The authors say "If ozone transported from outside Japan is not as reduced in future, efforts to reduce precursor emissions in Japan will not effectively contribute to the reduction in the concentrations of secondary PM2.5 components because OH that originated in ozone transported from outside Japan affects their formation", which is an interesting statement. But it is hard to figure out which sources are releasing PM2.5 precursors (for example NOx, SOx or VOCs) but not releasing O3 precursors. All combustion sources are strong VOC emitters, and efforts are made/have been made to abate NOx and SOx. Of course that the efforts in reducing emissions in Japan will not counteract the arrival of steady emissions from outside, but the reduction in precursor emissions in Japan will led to a lesser formation of secondary aerosols (although not in the same proportion as the applied reduction) and will contribute to the reduction of the continental O3 background.*

[Reply]

I agree your comments. Source releasing $PM_{2.5}$ precursors emit $O_3$ precursors. Discussions were too generalized. They should focus on $SO_4^{2-}$ and $NO_3^-$ as target species and $SO_2$ and $NO_X$ as precursors. Corresponding expressions in this paragraph in the lines 405-413 have been revised as follows.

"Section 3.2 discussed higher relative contributions than previous studies and less contrasts between western and eastern Japan for the sensitivities of $PM_{2.5}$ to s11 obtained in this study. Oxidation of $SO_2$ and $NO_X$ emitted from domestic sources by OH that originated in ozone transported from outside Japan is another factor that causes higher sensitivities of s11. The entirety of Japan is equally affected by ozone transported from outside Japan, as shown in Fig. 2(a), because of its long lifetime in the atmosphere, resulting in less contrast in the sensitivities of $PM_{2.5}$ to s11 between western and eastern Japan, whereas the sensitivities of domestic emissions are small. Ozone governs the oxidative capacity of the atmosphere (Prinn, 2003). If ozone transported from outside Japan is not as reduced in future, efforts to reduce $SO_2$ and $NO_X$ emissions in Japan will not effectively contribute to the reduction in the concentrations of $SO_4^{2-}$ and $NO_3^-$ because OH that originated in ozone transported from outside Japan affects their formation."

However, the sensitivities of ozone to domestic emissions are small. In addition, influences of emissions in Japan to background ozone are marginal. I think influences of emissions of ozone precursors in Japan on oxidation of $SO_2$ and $NO_2$ are limited. The following sentence has been inserted in the line 410.

"whereas the sensitivities of domestic emissions are small"

Dear Referee #3:

*[Referee #3]*

*General comment: technically sound but conclusions unclear and disappointing, possibly overstated*

*The authors have performed air quality simulations using the CMAQ model over various nested domains including Japan or parts of that country. They show in a convincing way that their simulations are realistic and have resonable (even good) performance. They study various methods to study the impacts of different types of sources in terms of concentrations of PM2.5 and ozone, including "Brute force method" (i.e. sensitivity simulations), ISAM and HDDM. As far as I can tell, all the methods implemented by the authors are technically sound and, at least in terms of modelled concentrations, they are easily on par with the State of the art.*

*However, even though this study has obviously involved a big amount of work, its point is not clear to me. In the abstract, the authors state that "This study demonstrated that a combination of sensitivities and apportionments derived by the BFM, HDDM, and ISAM can provide critical information to identify key emission sources and processes in the atmosphere, which are vital for the development of effective strategies for improved air quality". A similar statement appears in the conclusion: "This study demonstrated that a combination of sensitivities and apportionments derived by the BFM, HDDM, and ISAM can provide critical information to identify key emission sources and processes in the atmosphere, which are vital for the development of effective strategies for improved air quality, using consistent model configurations and inputs.". However, in-between I (and only "I" because that feeling is very possibly due to the fact that I am not so familiar with the issues the authors discuss) felt overwhelmed by a mass of plots and figures quite often lacking physico-chemical interpretation.*

*In summary, I have failed to understand which of the actual information unveiled by the authors was "critical" or even "vital" for policy design. On the contrary, I have the feeling that the methods they deploy are advanced but the actual results that they show are often disappointing when compared to the weaponry that they have used. For example, in the conclusion, the authors state that "Domestic sources had certain sensitivities to PM 2.5 , but significantly smaller or even negative sensitivities to ozone due to titration and nonlinear responses against precursor emissions.", which is hardly a surprise, it is discussed in all the good atmospheric composition textbooks that ozone concentrations are having a twofold sensitivity to emissions depending on the chemical regime. Here the authors' methodology seems to lead the reader to conclusions that are already very well-known.*

*I think the authors have realized good simulations of air quality over their areas of interest, convincingly shown that point, they have deployed methods they claim to be extremely useful in terms of understanding the rôle of different source areas and activity sectors in air pollution in Japan, but in my opinion they fail to make that second point, leading to disappointing conclusions.*

[Reply]

Thank you so much for critical comments. I also think that the works like this study have not be completed in any other previous studies. The results provided various interesting information. Indeed, nonlinear relationships between ambient concentrations of secondary pollutants including ozone and $PM_{2.5}$ and precursor emissions are well-known and written in textbooks. However, I believe that it is still worthwhile to investigate them further. As mentioned in the introduction, we are facing problems involving ozone and $PM_{2.5}$ in Japan in spite of stringent emission controls. That means our understandings on nonlinear relationships between concentrations and precursor emissions are not enough. Currently, we do not have any clear understandings for effectively suppressing concentrations of ozone and $PM_{2.5}$. We hope to contribute to solving the problems by providing useful scientific and quantitative information through this study. In addition, nonlinear relationships are not phenomena limited to Japan. We suppose our findings would be valuable in other countries and regions.

The paragraph has been be added at the end of Section 1 in the lines 85-91 as follows to explain our idea.

"There are well-known nonlinear relationships between ambient concentrations of secondary pollutants including ozone and secondary components involved in $PM_{2.5}$ (Seinfeld and Pandis, 1998). They are likely to cause deviations between source sensitivities and apportionments due to complex photochemical reactions and gas-aerosol partitioning. Nevertheless, it is important to investigate magnitudes of deviations and major causes of nonlinear relationships for considering effective strategies to suppress concentrations of secondary pollutants. Processes causing nonlinear relationships are universal phenomena and not limited to Japan. The findings of this study contribute not only to solving remaining issues involving ozone and $PM_{2.5}$ in Japan, but also to understanding on possible influences of nonlinear relationships in other countries and regions."

I have tried to explain the importance of this study throughout the manuscript. In addition, results and discussions of additional simulations, which were conducted based on the comments of another reviewer, have been added. Please check them in the revised manuscript. I hope these revisions are interesting for you and readers.

I have also revised the manuscript based on your comments below.

*[Referee #3]*
*Title:*
*I have a hard time understanding the title, "Comprehensive analyses of source sensitivities to and*

*apportionments of PM 2.5 and ozone over Japan via multiple numerical techniques". Even though it might be due to my partial knowledge of the jargon in this particular field, I have the feeling that, in the title and the rest of the text (e.g. l. 55, l. 74 and following, etc.). It seems that in the author's vocabulary they adress the sensitivity of the NOx emissions to ozone concentrations (this is just an example) while the ordinary way of thinking is more to assess the sensitivity of ozone concentrations to NOx emissions.*

[Reply]

I am so sorry for this English mistake for the important words of this study. Another referee raised the same issue. "Sensitivity of pollutant concentrations to emissions" and "apportionment of pollutant concentrations to emissions" should be correct expressions.

The title has been changed as follows.

"Comprehensive analyses of source sensitivities and apportionments of $PM_{2.5}$ and ozone over Japan via multiple numerical techniques"

In addition, I have checked the main text, tables, figures, and supplemental material, to correct all the relevant parts. A grammar check has been also done again. Please check in the revised manuscript.

*[Referee #3]*

*Major comments:*

*l. 461-464: "While PM 2.5 concentrations and their absolute sensitivities of all the sources were lower than those calculated by previous studies for past years due to emission reductions, the relative contributions of transport from outside Japan to the total sensitivities were even larger, suggesting that emissions in Japan have been reduced similar to surrounding countries, including China." I think the sensitivities and apportionment calculated by the authors do not depend on the actual emissions by Japan and China but on the emission hypotheses and inventories that have been chosen by the authors. I do not think the authors can draw any conclusion from their study regarding the emission reduction paths followed by Japan or China. I think the logical path leading to this result is circular: the authors make certain choices regarding emissions in Japan and China, they observe that the results they obtain are consistent with the hypothesis they made, but in my opinion this is no proof that their initial hypothesis is correct.*

[Reply]

I admit that this is very important issue. However, emissions compiled in the emission inventory have

been estimated based on various information including changes in energy consumption, emission factors, and implementation of emission controls. Every simulation study must rely on one of emission inventories as a first assumption. Agreement of observed and simulated concentrations could be considered as one of proofs for accuracy of the emission inventory. However, it is indeed impossible to conclude only from this fact that the emission inventory is definitely accurate. Circular exercises including validation of simulations and improvement of emission inventories are necessary. Regarding this study, while simulations implied that emissions in Japan have been reduced as estimated in emission inventories, they also implied reductions may be too much and caused underestimation of $PM_{2.5}$ concentrations. Not only the former but also the latter aspects are discussed in the lines 265-269. Discussions for the latter aspect have been revised to make clearer as follows.

"However, we can also state that the underestimations of the $PM_{2.5}$ concentrations are larger in eastern than western Japan as described in section 3.1. Influences of domestic sources should be accumulated more in eastern than western Japan because the prevalent air flow over Japan is westerly. Therefore, worse model performance in eastern Japan imply underestimation of domestic emissions. Reductions of domestic emissions from fiscal years 2005 to 2015 may be overestimated."

*[Referee #3]*
*Minor comments, typos :*
*p. 1, l. 16-17: "While domestic sources had certain source apportionments to ozone concentrations, transport from outside Japan dominated the source sensitivities." If possible, many sentences of this kind should be formulated in a more intuitive way, e.g., while domestic sources can contribute to a certain extent to simulated ozone concentrations, transport from outside Japan can be considered as the main overall driver of ozone concentrations in Japan (this is only my interpretation of course, just as an example on how the authors should make their conclusions more accessible to readers in the field but not specialized). At all places where this is possible, the authors should formulate their statements and partial conclusions in more physical terms.*

[Reply]
While it is a bit difficult to revise as suggested because sensitivities and apportionments should be clearly distinguished in this study, I have tried to make descriptions in a more intuitive way. Please check them in the revised manuscript.

*[Referee #3]*
*p. 1, l. 22: "that that"*

[Reply]

I am sorry for this mistake. It has been corrected.

*[Referee #3]*
*l. 96: "Following" seems useless.*

[Reply]

It has been removed.

Reference

[revised manuscript text omitted]

**Supplementary material**

**Table S1: Annual total emission amounts (Mg/year) of each source group for the 2016 fiscal year in d02.**

| Group | CO | SO$_2$ | NO$_X$ | NH$_3$ | NMVOC[*1] | PM$_{2.5}$ | EC[*2] | OC[*3] |
|---|---|---|---|---|---|---|---|---|
| s01 | 1,171,555 | 910 | 411,047 | 15,576 | 122,712 | 23,850 | 6,091 | 5,361 |
| s02 | 59,862 | 279,102 | 682,708 | 0 | 20,593 | 55,988 | 10,766 | 16,045 |
| s03 | 164,110 | 3,057 | 97,754 | 0 | 11,404 | 4,479 | 2,503 | 1,421 |
| s04 | 1,230,684 | 356,649 | 761,388 | 0 | 226,562 | 19,196 | 1,661 | 3,459 |
| s05 | 69,836 | 369 | 4,924 | 943 | 8,016 | 10,028 | 586 | 6,997 |
| s06 | 46,034 | 840 | 33,421 | 0 | 11,938 | 1,770 | 190 | 619 |
| s07 | 0 | 0 | 0 | 0 | 587,827 | 0 | 0 | 0 |
| s08 | 0 | 0 | 0 | 389,915 | 55,714 | 1,811 | 0 | 0 |
| s09[*4] | 404,322 | 1,534,513 | 58,700 | 21,732 | 3,954,064 | 0 | 0 | 0 |
|  | 132,359 | 1,534,513 | 9,940 | 21,732 | 1,302,798 | 0 | 0 | 0 |
| s10 | 13,152,939 | 1,168,210 | 2,990,211 | 649,985 | 2,108,940 | 736,968 | 112,112 | 241,057 |

[*1] Non-methane volatile organic compound
[*2] Elemental carbon
[*3] Organic carbon
[*4] Lower values indicate amounts within Japan only

**Table S2: Statistics of the model performance on the MDA8O3 and daily mean PM$_{2.5}$ concentrations for the entire 2016 fiscal year 2016 in the regions.**

| Species | Region | Number[1] | Obs.[2] | Sim.[3] | NMB[4] | NME[5] | R[6] |
|---|---|---|---|---|---|---|---|
| MDA8O3 (ppb) | JP | 1150 | 42.7 | 46.0 | 7.79% | 20.0% | 0.860 |
| | KO | 151 | 43.4 | 48.4 | 11.3% | 24.0% | 0.820 |
| | CS | 143 | 44.4 | 46.8 | 5.27% | 20.7% | 0.834 |
| | KS | 176 | 43.1 | 46.2 | 7.14% | 20.0% | 0.868 |
| | TH | 201 | 43.6 | 47.4 | 8.69% | 18.9% | 0.869 |
| | KK | 369 | 41.5 | 44.3 | 6.94% | 19.5% | 0.872 |
| | HT | 110 | 39.4 | 44.1 | 11.7% | 21.2% | 0.797 |
| | OH | 108 | 42.6 | 45.2 | 6.10% | 20.7% | 0.864 |
| | AM | 75 | 42.6 | 45.6 | 6.89% | 19.3% | 0.874 |
| | ST | 196 | 40.8 | 42.9 | 5.21% | 19.7% | 0.880 |
| PM$_{2.5}$ (µg/m$^3$) | JP | 820 | 11.9 | 7.62 | -35.9% | 41.6% | 0.852 |
| | KO | 127 | 14.2 | 10.3 | -27.2% | 36.6% | 0.860 |
| | CS | 113 | 13.6 | 9.26 | -30.3% | 38.9% | 0.853 |
| | KS | 134 | 12.0 | 7.77 | -35.5% | 39.5% | 0.862 |
| | TH | 132 | 10.7 | 6.62 | -38.2% | 42.0% | 0.855 |
| | KK | 243 | 11.3 | 6.48 | -42.8% | 46.2% | 0.836 |
| | HT | 71 | 9.01 | 5.41 | -39.9% | 46.0% | 0.827 |
| | OH | 71 | 12.5 | 8.30 | -33.8% | 38.2% | 0.863 |
| | AM | 43 | 11.6 | 7.40 | -36.3% | 40.0% | 0.855 |
| | ST | 156 | 12.0 | 6.68 | -44.5% | 46.8% | 0.839 |
| SO$_4^{2-}$ (µg/m$^3$) | JP | 154 | 2.73 | 2.64 | -3.29% | 40.9% | 0.710 |
| NO$_3^-$ (µg/m$^3$) | JP | 154 | 0.641 | 1.01 | 57.1% | 121% | 0.441 |
| NH$_4^+$ (µg/m$^3$) | JP | 154 | 1.11 | 1.08 | -3.07% | 41.1% | 0.704 |
| EC (µg/m$^3$) | JP | 136 | 0.757 | 0.345 | -54.4% | 58.3% | 0.477 |
| OC (µg/m$^3$) | JP | 151 | 2.58 | 0.958 | -62.8% | 66.0% | 0.487 |

[1] Number of monitoring stations
[2] Observed values
[3] Simulated values
[4] Normalized Mean Bias
[5] Normalized Mean Error
[6] Correlation coefficient

**Table S3: Source sensitivities  of the annual mean ozone and PM2.5 concentrations simulated in the regions. The upper table shows ratios (%) against the simulated concentrations. The lower table shows their normalized ratios (%).**

(a1) O₃ (not normalized)

| Group | JP | KO | CS | KS | TH | KK | HT | OH | AM | ST |
|-------|------|------|------|------|------|------|------|------|------|------|
| s01 | -0.8 | -1.0 | -0.5 | -1.6 | -1.1 | -1.7 | -0.1 | -5.2 | -5.9 | -8.0 |
| s02 | 0.6 | 0.7 | -0.1 | 0.5 | 0.8 | 0.7 | 0.8 | -1.7 | -0.3 | -0.5 |
| s03 | -0.2 | -0.1 | -0.1 | -0.4 | -0.3 | -0.5 | 0.0 | -1.7 | -1.7 | -2.5 |
| s04 | -0.6 | -0.7 | -1.2 | -2.0 | -0.6 | -1.2 | 0.4 | -8.3 | -7.8 | -9.6 |
| s05 | 0.0 | 0.0 | 0.0 | 0.0 | 0.0 | 0.0 | 0.0 | 0.0 | 0.0 | 0.0 |
| s06 | -0.2 | -0.1 | -0.1 | -0.3 | -0.2 | -0.3 | -0.1 | -1.0 | -0.8 | -1.6 |
| s07 | 0.6 | 0.5 | 0.7 | 0.9 | 1.0 | 1.1 | 0.2 | 1.5 | 1.8 | 2.3 |
| s08 | 0.0 | -0.1 | 0.0 | 0.0 | 0.0 | -0.1 | 0.0 | 0.0 | 0.0 | -0.1 |
| s09 | 1.0 | 1.2 | 1.5 | 1.6 | 1.4 | 1.2 | 0.4 | 2.3 | 2.0 | 1.9 |
| s10 | 1.1 | 0.2 | 0.6 | 1.1 | 1.4 | 1.5 | 1.3 | 1.2 | 1.5 | 1.5 |
| s11 | 77.5 | 77.7 | 75.1 | 75.1 | 73.3 | 75.9 | 81.0 | 81.7 | 80.0 | 85.3 |
| s12 | -0.2 | -0.2 | -0.2 | -0.2 | -0.2 | -0.2 | -0.1 | -0.3 | -0.3 | -0.3 |
| Sum | 79.0 | 78.0 | 75.6 | 74.6 | 75.4 | 76.3 | 83.8 | 68.4 | 68.5 | 68.4 |

(a2) O₃ (normalized)

| Group | JP | KO | CS | KS | TH | KK | HT | OH | AM | ST |
|-------|------|------|------|------|------|------|------|------|------|------|
| s01 | -1.0 | -1.3 | -0.7 | -2.1 | -1.5 | -2.3 | -0.1 | -7.6 | -8.6 | -11.7 |
| s02 | 0.8 | 0.9 | -0.1 | 0.7 | 1.1 | 0.9 | 1.0 | -2.5 | -0.4 | -0.8 |
| s03 | -0.2 | -0.2 | -0.1 | -0.6 | -0.4 | -0.6 | 0.0 | -2.5 | -2.5 | -3.6 |
| s04 | -0.7 | -0.9 | -1.6 | -2.7 | -0.9 | -1.6 | 0.4 | -12.1 | -11.4 | -14.0 |
| s05 | 0.0 | 0.0 | 0.0 | 0.0 | 0.0 | 0.0 | 0.0 | 0.0 | 0.0 | 0.0 |
| s06 | -0.2 | -0.1 | -0.1 | -0.4 | -0.2 | -0.4 | -0.1 | -1.5 | -1.1 | -2.4 |
| s07 | 0.8 | 0.6 | 0.9 | 1.2 | 1.3 | 1.4 | 0.2 | 2.2 | 2.6 | 3.3 |
| s08 | 0.0 | -0.1 | 0.0 | 0.0 | 0.0 | -0.1 | 0.0 | 0.0 | 0.0 | -0.1 |
| s09 | 1.3 | 1.5 | 2.0 | 2.1 | 1.8 | 1.6 | 0.5 | 3.4 | 2.9 | 2.7 |
| s10 | 1.4 | 0.3 | 0.8 | 1.4 | 1.9 | 1.9 | 1.5 | 1.7 | 2.2 | 2.2 |
| s11 | 98.2 | 99.6 | 99.3 | 100.6 | 97.1 | 99.5 | 96.6 | 119.4 | 116.9 | 124.7 |
| s12 | -0.2 | -0.3 | -0.3 | -0.3 | -0.3 | -0.3 | -0.2 | -0.4 | -0.4 | -0.4 |

**Table S3:** *Cont'd.*

(b1) PM$_{2.5}$ (not normalized)

| Group | JP | KO | CS | KS | TH | KK | HT | OH | AM | ST |
|---|---|---|---|---|---|---|---|---|---|---|
| s01 | 4.6 | 3.8 | 3.2 | 5.5 | 6.1 | 8.3 | 3.1 | 9.1 | 10.6 | 12.6 |
| s02 | 5.9 | 5.9 | 6.9 | 7.8 | 7.5 | 6.2 | 4.0 | 10.1 | 10.7 | 9.4 |
| s03 | 0.7 | 0.5 | 0.4 | 0.8 | 1.0 | 1.5 | 0.5 | 1.5 | 1.8 | 2.6 |
| s04 | 8.2 | 6.0 | 8.0 | 9.4 | 11.0 | 12.2 | 6.6 | 10.5 | 13.9 | 14.2 |
| s05 | 0.9 | 0.6 | 0.4 | 1.0 | 1.0 | 1.7 | 1.0 | 2.0 | 2.0 | 3.8 |
| s06 | 0.3 | 0.2 | 0.1 | 0.2 | 0.3 | 0.6 | 0.4 | 0.3 | 0.5 | 0.8 |
| s07 | 0.3 | 0.1 | 0.3 | 0.4 | 0.5 | 0.9 | 0.1 | 0.7 | 1.0 | 2.0 |
| s08 | 10.8 | 12.5 | 10.7 | 10.4 | 10.6 | 13.6 | 8.5 | 10.0 | 13.3 | 13.5 |
| s09 | 7.6 | 11.7 | 9.1 | 8.4 | 7.9 | 7.6 | 3.9 | 7.1 | 6.1 | 5.4 |
| s10 | 11.6 | 11.6 | 13.2 | 11.9 | 11.0 | 9.6 | 11.8 | 10.7 | 9.6 | 7.2 |
| s11 | 66.2 | 65.9 | 65.4 | 61.8 | 60.4 | 59.7 | 73.1 | 56.8 | 55.6 | 56.4 |
| s12 | -3.9 | -4.7 | -4.4 | -4.4 | -4.3 | -4.2 | -2.8 | -4.3 | -5.4 | -4.4 |
| Sum | 113.2 | 114.0 | 113.3 | 113.4 | 113.0 | 117.6 | 110.3 | 114.5 | 119.6 | 123.4 |

(b2) PM$_{2.5}$ (normalized)

| Group | JP | KO | CS | KS | TH | KK | HT | OH | AM | ST |
|---|---|---|---|---|---|---|---|---|---|---|
| s01 | 4.0 | 3.4 | 2.8 | 4.9 | 5.4 | 7.1 | 2.8 | 8.0 | 8.8 | 10.2 |
| s02 | 5.2 | 5.2 | 6.1 | 6.9 | 6.6 | 5.2 | 3.6 | 8.8 | 8.9 | 7.6 |
| s03 | 0.6 | 0.4 | 0.3 | 0.7 | 0.9 | 1.3 | 0.4 | 1.3 | 1.5 | 2.1 |
| s04 | 7.3 | 5.2 | 7.0 | 8.3 | 9.7 | 10.4 | 6.0 | 9.1 | 11.6 | 11.5 |
| s05 | 0.8 | 0.5 | 0.4 | 0.9 | 0.9 | 1.4 | 0.9 | 1.7 | 1.7 | 3.1 |
| s06 | 0.3 | 0.1 | 0.1 | 0.2 | 0.3 | 0.5 | 0.4 | 0.2 | 0.4 | 0.7 |
| s07 | 0.3 | 0.1 | 0.2 | 0.4 | 0.5 | 0.8 | 0.1 | 0.6 | 0.8 | 1.6 |
| s08 | 9.5 | 11.0 | 9.5 | 9.2 | 9.4 | 11.6 | 7.7 | 8.8 | 11.1 | 10.9 |
| s09 | 6.7 | 10.2 | 8.0 | 7.4 | 7.0 | 6.4 | 3.6 | 6.2 | 5.1 | 4.4 |
| s10 | 10.3 | 10.2 | 11.7 | 10.5 | 9.7 | 8.1 | 10.7 | 9.4 | 8.1 | 5.9 |
| s11 | 58.5 | 57.8 | 57.7 | 54.5 | 53.4 | 50.8 | 66.3 | 49.7 | 46.5 | 45.7 |
| s12 | -3.5 | -4.1 | -3.9 | -3.8 | -3.8 | -3.6 | -2.5 | -3.7 | -4.5 | -3.6 |

[Figure]

**(a) MDA8O3**

**Figure S1: Comparisons of the observed and simulated monthly mean MDA8O3 and PM$_{2.5}$ concentrations at all stations in the regions. Markers and error bars represent mean values and standard deviations, respectively, of the daily concentrations at all monitoring stations for each month.**

[Figure]

**(b) PM$_{2.5}$**

**Figure S1:** *Cont'd*

[Figure]

**Figure S2: Scatter plots of the observed and simulated concentrations of PM2.5 components during the monitoring campaigns of ambient concentrations of PM2.5 components at all the locations throughout Japan in all four seasons. Regression lines are represented by red lines.**

[Figure]

**Figure S3:** Source sensitivities  of the annual mean concentrations of PM2.5 components derived by BFM in the regions. Thick black lines represent the simulated concentrations.

[Figure]

**Figure S4: Source sensitivities  of the monthly mean concentrations of PM2.5 components derived by BFM in entire Japan (JP) and ST. Thick black lines represent the simulated concentrations.**

[Figure]

**Figure S4:** *Cont'd*

[Figure]

**Figure S5: Apportionments derived by ISAM and sensitivities derived by BFM and HDDM of  the simulated concentrations of PM2.5 components to all source groups in JP and ST for the two target weeks in the four seasons.**

[Figure]

**Figure S5:** *Cont'd*

[Figure]

**Figure S6: Horizontal distributions of the apportionments and the sensitivities  to s11 for the target two weeks of the spring.**

[Figure]

**Figure S7: Sensitivities derived by BFM-20, BFM-100, HDDM-20, and HDDM-100, and apportionments derived by ISAM of the daily NO$_3^-$ and NH$_4^+$ concentrations to s04 and s08 for the two target weeks in winter in ST.**

[Figure]

[Figure]

**Figure** **S8**: **Sensitivities derived by BFM-20 (left) and BFM-100 (middle), and a**pportionments **derived by ISAM**  **right) of** **the hourly ozone concentrations (shown by a line with markers) to** NO_X **and VOC emissions of s01**   **on July 25th in ST.**

[Figure]

**Figure S9: Horizontal distributions of the apportionments and sensitivities of ₓ the ozone concentrations to the s01 NOₓ and VOC emissions averaged for the two target weeks in summer.**

[Figure]

**Figure s10.** Source sensitivities of the annual mean PM$_{2.5}$ concentrations derived by BFM in the regions. Thick black lines represent the simulated concentrations. The values shown in (b) were scaled by ratios of observed and simulated concentrations of PM$_{2.5}$ components.

---

## Author Response (AR2)

Dear Editor:

We really appreciate your acceptance of our paper. The manuscript has been slightly revised as described in the response to the referee #3.

Dear Referee #3:

*[Referee #1]*

*I have appreciated the authors' efforts in improving the manuscript's clarity. I suggest publication of this manuscript after some technical corrections are brought.*

[Reply]

Thank you so much for your review and understandings on our paper. The manuscript has been slightly revised as described below.

*[Referee #1]*

*"these are vital for the development of effective strategies for improved air quality, using consistent model configurations and inputs. However, model configurations and inputs may not necessarily be consistent.", it is not really clear what the authors mean in these lines.*

[Reply]

This sentence has been simplified and revised as follows.

"The sensitivities and apportionments were derived with the consistent model configurations and inputs in this study."

*[Referee #1]*

*"which are clearly reflected in their sensitivities of $SO_4^{2-}$" I think here "to" is meant rather than "off".*

[Reply]

This part has been revised as follows.

[revised manuscript text omitted]